# AD-HOC HUMAN-AI COORDINATION CHALLENGE

## ABSTRACT

Achieving seamless coordination between AI agents and humans is crucial for real-world applications, yet it remains a significant open challenge. Hanabi is an established, fully cooperative benchmark environment that involves imperfect information, limited communication, theory of mind, and the necessity for coordination among agents to achieve a shared goal. These characteristics, in principle, make Hanabi an ideal testbed for exploring human-AI coordination. However, one key issue is that evaluation with human partners is both expensive and difficult to reproduce. To address this, we first develop *human proxy agents* via a combination of behavioural cloning on a large-scale dataset of human game play and regularised reinforcement learning. These proxies serve as robust, cheap and reproducible human-like evaluation partners in our Ad-Hoc Human-AI Coordination Challenge (AH2AC2). To facilitate the exploration of methods that leverage *limited amounts* of human data, we introduce a data-limited challenge setting, using 1,000 games, which we open-source. Finally, we present baseline results for both two-player and three-player Hanabi scenarios. These include zero-shot coordination methods, which do not utilise any human data, and methods that make use of the available human data combined with reinforcement learning. To prevent overfitting and ensure fair evaluation, we introduce an evaluation protocol that involves us hosting the proxy agents rather than publicly releasing them, and a public leaderboard for tracking the progress of the community. We make our code available as an anonymous repository: https://anonymous.4open.science/r/ah2ac2-E451/

## 1 INTRODUCTION

The interaction between humans and artificial intelligence (AI) systems is rapidly evolving, driven by significant AI progress and the increasing integration of AI into various aspects of our lives (Rawas, 2024). As AI agents become more sophisticated, capable and prevalent, effective coordination between humans and AI becomes crucial. This interaction spans a wide range of domains, from collaborative decision-making in healthcare (Asan et al., 2020) to shared control in autonomous vehicles (Bansal et al., 2018) and robotics (Haarnoja et al., 2018; Ahn et al., 2024) as well advanced digital assistants (Gemini Team, 2024; Bai et al., 2022). The ultimate goal is to create AI agents that are not limited to solving problems independently but can also work effectively with humans to complete tasks in human-compatible ways (Russell, 2019; Carroll et al., 2019).

Traditional approaches to training AI agents frequently employ *self-play* (SP), where agents learn by interacting with copies of themselves (Tesauro, 1994). While SP has been successful in competitive games like chess and Go (Silver et al., 2016), this approach can lead to agents that overfit to specific strategies, limiting their ability to generalise to novel partners (Carroll et al., 2019). In human-AI coordination scenarios, forming rigid conventions is particularly problematic, as doing so can pose safety risks and humans may be unable to adapt appropriately (Bard et al., 2019). Instead, AI agents must adapt to human partners, taking into account their different skill levels, preferences, and ways of communicating (Hu et al., 2021; 2022).

Hanabi is an established, fully cooperative benchmark environment that involves imperfect information, limited communication, theory of mind, and the necessity for coordination among different players to achieve a shared goal (Bard et al., 2019). These characteristics, in principle, make Hanabi an ideal testbed for evaluating human-AI coordination. However, the evaluation of AI agents when paired with human partners is both expensive and difficult to reproduce. Therefore, many previ-

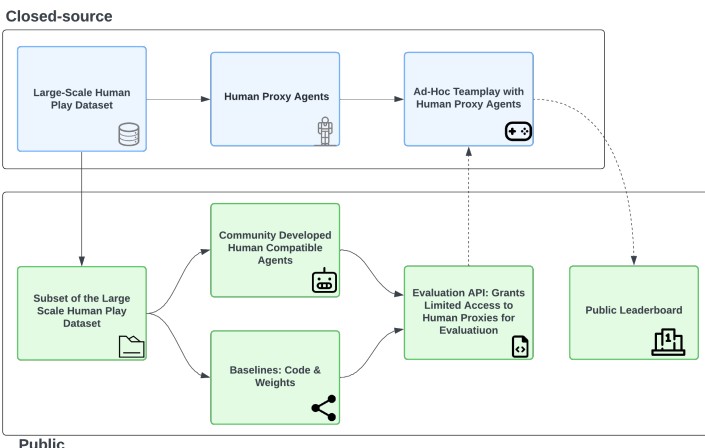

Figure 1: Ad-Hoc Human-AI Coordination Challenge (AH2AC2)

ous research efforts in Hanabi have utilised held-out sets of human data and human proxy agents to evaluate human-AI coordination (Hu et al., 2022; 2021; Lupu et al., 2021; Jacob et al., 2022; Bakhtin et al., 2022). Yet, human gameplay datasets and the human proxy agents derived from these datasets have thus far remained closed-source and unavailable to the wider research community. To address these problems, we introduce the *Ad-Hoc Human-AI Coordination Challenge* (AH2AC2) as a standardised way to evaluate AI agents' ability to collaborate with human-like partners in Hanabi. To create realistic human-like evaluation partners, we develop human proxy agents using a combination of behavioural cloning (BC) and regularised reinforcement learning (RL). The BC component leverages a large-scale dataset of human gameplay, comprising 101,096 two-player games and 46,525 three-player games. Subsequently, we refine the BC policy using Independent Proximal Policy Optimisation (IPPO) (de Witt et al., 2020; Schulman et al., 2017) augmented with a regularisation term. This regularisation encourages the learned policy to adhere closely to human play styles (Bakhtin et al., 2022; Hu et al., 2022; Cornelisse & Vinitsky, 2024). Through empirical evaluation, we demonstrate that these proxy agents outperform pure imitation learning while exhibiting behaviour consistent with human players. Our human proxies serve as robust, cheap and reproducible evaluation partners in AH2AC2, enabling consistent assessment of AI agents' ability to coordinate with human-like players.

To ensure evaluation integrity and prevent overfitting, we restrict public access to the human proxy agents and their large-scale training dataset. Instead, we introduce a data-limited challenge to facilitate research using limited human data - using 1,000 games for two- and three-player settings which we open-source. Finally, we provide baselines for each challenge variety. These include zero-shot coordination methods, which do not utilise human data, such as Off-Belief Learning (OBL) (Hu et al., 2021), as well as methods that use the available human data, including BC and best response to behavioural cloning policy (BR-BC) (Carroll et al., 2019).

The summary of our key contributions is as follows:

- We develop four human proxy agents (two for the two-player setting and two for the three-player setting) using a combination of BC on a large-scale human play dataset and regularised RL. These agents form the basis of the AH2AC2 evaluation procedure.

- We release the first open-source Hanabi human play dataset - featuring 1,858 two-player and 1,221 three-player games.

- We establish the AH2AC2, featuring two sub-challenges where participants are allowed to utilise limited amounts of human play data - 1,000 games we open-source - during agent training.

- We create an evaluation protocol where we host the human proxy agents rather than publicly releasing them. We maintain a public leaderboard for tracking progress.

- We provide a diverse set of baselines for two- and three- player settings. We include methods that utilise human data and those that do not.

## 2 BACKGROUND

### 2.1 HANABI

Hanabi is a cooperative card game designed for 2-5 players, but we restrict ourselves to two- and three- player settings in this work due to the availability of human gameplay data for these configurations. In Hanabi, players aim to collectively build five ascending stacks of cards, one for each of the five colours. The deck comprises 50 cards, with 10 cards of each colour. These include three cards of rank 1, two each of ranks 2, 3, and 4, and a single card of rank 5. A unique feature of the game is that players can see each other's cards, but not their own. Information about one's own cards is gained through hints from teammates or by interpreting their actions.

The team starts with eight information tokens and three lives. On their turn, players can choose one of the following three actions: (1) A player can choose a teammate and provide information about all the cards of a specific colour or rank in that teammate's hand. The hint must be complete, meaning all cards of the specified colour or rank must be indicated. Each hint consumes one of the limited information tokens. If no information tokens remain, this action is unavailable. (2) If the team has fewer than eight information tokens, a player can discard a card face-up, revealing it to all players. This action replenishes one information token and the player draws a new card. (3) A player can attempt to play a card from their hand onto the corresponding colour stack. If the card's rank is the next in the sequence for that colour, it is successfully played. Successfully playing a rank five card earns the team an additional information token. However, an incorrect play results in the loss of one of the team's limited lives. Regardless of the outcome, the player draws a new card from the deck.

The game concludes when one of the following three conditions is met: (1) the team loses all their lives, (2) all five stacks are completed, or (3) the deck is depleted. The final score reflects the team's success in building the stacks. The maximum score is 25, and it is determined by summing up all the played cards. When all lives are lost, the score is zero.

### 2.2 DEC-POMDP

We consider a decentralised partially observable Markov decision process (Dec-POMDP) (Oliehoek, 2012), defined as a tuple $\mathcal{M} = (n, \mathcal{S}, \mathcal{A}, P, r, O, \mathcal{O}, \gamma, T)$, where $n$ is the number of agents, $\mathcal{S}$ is the finite set of states, and $\mathcal{A} = \times_{j=1}^{n} \mathcal{A}^i$ is the joint-action space with $\mathcal{A}^i$ being the action space for agent $i$. The transition function $P : \mathcal{S} \times \mathcal{A} \times \mathcal{S} \to [0, 1]$ defines the probability of transitioning to state $s_{t+1}$ given state $s_t$ and joint action $a_t$. The reward function is $r : \mathcal{S} \times \mathcal{A} \to \mathbb{R}$, the observation function is $O : \mathcal{O} \times \mathcal{S} \to [0, 1]$, where $\mathcal{O} = \times_{i=1}^{n} \mathcal{O}^i$ is the joint observation space, and $\gamma \in [0, 1]$ is the discount factor, with $T$ being the horizon. At time $t$, the Dec-POMDP is in state $s_t \in \mathcal{S}$ and generates a stochastic joint observation $o_t = (o_t^1, \ldots, o_t^n)$ sampled from $O(\cdot|s_t)$.

The trajectory up to time $t$ is defined as $\tau_t = (o_0, a_0, \ldots, o_{t-1}, a_{t-1}, o_t)$. We also define the action-observation history (AOH) of player $i$ as $\tau_t^i = (o_0^i, a_0^i, \ldots, o_{t-1}^i, a_{t-1}^i, o_t^i)$. Each player $i$ selects action according to a policy $\pi^i(a_t^i|\tau_t^i)$, which maps AOH to a probability distribution over actions. The joint policy $\pi$ is composed of these individual policies, resulting in a joint action $a_t = (a_t^1, \ldots, a_t^n) \in \mathcal{A}$ selected with probability $\pi(a_t|\tau_t) = \prod_{j=1}^{n} \pi^i(a_t^i|\tau_t^i)$. The environment then transitions to state $s_{t+1}$ according to $P(s_{t+1}|s_t, a_t)$, and all agents receive the shared reward $r(s_t, a_t)$. Considering the discount factor $\gamma$, the discounted return is defined as $R(\tau) = \sum_{t=0}^{T} \gamma^t r(s_t, a_t)$.

### 2.3 ZERO-SHOT COORDINATION AND AD-HOC TEAMPLAY

In many real-world scenarios, AI agents need to coordinate with unknown partners, including humans. Traditional cooperative multi-agent RL often uses SP, where agents train together to maximise joint returns (Bard et al., 2019; Carroll et al., 2019). However, SP training can lead to agents developing arbitrary conventions that work only with their training partners, failing when paired with independently trained agents—a critical issue for human-AI coordination (Carroll et al., 2019; Strouse et al., 2021). Zero-shot coordination (ZSC) addresses this challenge by training agents to coordinate effectively with new partners using the same algorithm (Hu et al., 2020; Treutlein et al., 2021). In ZSC, each agent independently trains a joint policy using a general-purpose learning al-

gorithm. After training, agents from different runs are paired, and their performance is evaluated through cross-play. The objective is to maximise the expected return when independently trained agents are matched. Ad-hoc teamplay focuses on an agent's ability to cooperate with unfamiliar teammates at test time (Stone et al., 2010). Similar to ZSC, agents are evaluated based on their performance with partners they have not trained with. However, unlike ZSC, ad-hoc teamplay does not require these teammates to utilise the same training algorithm. In this work, we are specifically interested in the scenario of ad-hoc teamplay with human and human-like agents.

## 3 RELATED WORK

While Hanabi was introduced as a benchmark encompassing both SP and ad-hoc teamplay (Bard et al., 2019), significant progress has been primarily confined to SP, with methods like SPARTA (Lerer et al., 2019) achieving near-perfect 24.61/25 points on average. However, agents trained in SP often rely on specialised conventions, leading to poor generalisation when paired with novel teammates. Consequently, ad-hoc teamplay, especially with humans, presents a more demanding and unsolved challenge.

A central issue when evaluating agent abilities for ad-hoc coordination is the selection of test-time policies. One approach to tackle this issue is to ensure the diversity of policies, thereby minimising the possibility of favouring any specific agent. For example, Cui et al. (2023) introduce ADVER-SITY that aims to produce highly skilled and reasonable policies that play according to diverse conventions. Another approach is to focus on a set of policies that hold intrinsic value, with the most natural choice being human policies. Coordinating with humans can be seen as a specialised form of ad-hoc team play, where the set of test policies comprises human strategies, which are inherently valuable due to their real-world relevance.

Ad-hoc human-AI coordination in Hanabi is explored in many previous works, with each presenting a different methodology for acquiring human proxy agents and evaluating human-AI coordination capabilities (Hu et al., 2021; 2020; Cui et al., 2023). However, there is no standard approach for ad-hoc human-AI coordination evaluation in the existing literature. Therefore, our work addresses this gap and proposes the AH2AC2.

Recent works have empirically shown that augmenting imitation learning methods with regularised RL creates stronger and more reliable policies. Multiple works have shown that regularised RL leads to policies that are more compatible with existing social conventions of the human reference group (Jacob et al., 2022; Bakhtin et al., 2022; Hu et al., 2022). Recently, (Cornelisse & Vinitsky, 2024) extended these works to the driving setting, where the authors show that data-driven regularisation leads to human-compatible policies. Our work builds upon these findings to build strong and human-like human proxy agents that enable AH2AC2.

Nekoei et al. (2023) motivates the multi-agent RL community to shift focus towards the few-shot adaptation problem, alongside the well-studied ZSC problem. The authors provide empirical evidence highlighting the limitations of current state-of-the-art ZSC algorithms in adapting to new partners. By introducing data limit settings, we aim to tie AH2AC2 and few-shot coordination. However, it is crucial to note a key distinction, in our setting, the few-shot-like effort is offline and one-sided. Human proxy agents do not adapt during test time, maintaining fixed behaviour. Instead, we allow for the usage of a small sample of the human play data that we used for training human proxy agents. This contrast underscores the unique challenges of coordinating with human-like agents, where adaptation must occur primarily on the part of the AI agent, given the constraints of human behaviour.

## 4 AD-HOC HUMAN-AI COORDINATION CHALLENGE (AH2AC2)

### 4.1 OVERVIEW

We collected a large-scale dataset from the hanab.live platform, consisting of 101,096 two-player games and 46,525 three-player games. This dataset, which is not publicly available, shows higher average scores compared to previously used (closed-source) datasets (Hu et al., 2021; 2022; Bakhtin

et al., 2022). We attribute this use of "H-group conventions", known for their effectiveness in Hanabi. More information about the large-scale dataset is available in the Appendix A.4.

Using the entire dataset, we develop human proxies that act as standard and cheap test partners for ad-hoc human-AI coordination evaluation in AH2AC2. These agents are trained on the entire dataset, which we plan to open-source after the challenge concludes. To prevent overfitting, we created an evaluation protocol where we host the human proxies instead of releasing them publicly.

As part of the AH2AC2 foundation, we release 3,079 games from the larger dataset — 1,858 two-player and 1,221 three-player games. Participants can use up to 1,000 games for model training. For validation, we provide a split containing 858 two-player and 221 three-player games, ensuring all starting decks in the validation set are unique. We summarise key statistics for our data split in Table 1.

During evaluation, participants' candidate agents are strictly limited to 1,000 evaluation games with our human proxies - no additional games are allowed. This controlled access ensures consistency across submissions. The candidate agent's performance is evaluated based on the mean and median scores achieved across 1,000 games with human proxies. For completeness, we also assess the agent's ability to predict human actions in an unseen set of human-played games.

Table 1: Statistics for open-sourced data.

| Setting | Data Budget | Metric | Min | Max | Avg | Median | Std |
|---------|-------------|--------|-----|-----|-----|--------|-----|
| Two-Player | 1,000 Games | Scores | 13 | 25 | 23.10 | 24 | 2.09 |
| | | Game Lengths | 53 | 76 | 65.75 | 66 | 3.59 |
| | Validation | Scores | 17 | 25 | 23.68 | 24 | 1.50 |
| | | Game Lengths | 52 | 73 | 65.09 | 65 | 3.02 |
| Three-Player | 1,000 Games | Scores | 14 | 25 | 23.05 | 24 | 1.97 |
| | | Game Lengths | 45 | 67 | 58.24 | 59 | 3.37 |
| | Validation | Scores | 19 | 25 | 24.19 | 25 | 1.20 |
| | | Game Lengths | 46 | 62 | 56.16 | 56 | 2.86 |

## 4.2 EVALUATION PROTOCOL

### HUMAN PROXIES

We develop four human proxy agents for evaluating ad-hoc human-AI coordination: two for two-player Hanabi and two for three-player. These agents are trained using HDR-IPPO, a procedure combining BC and regularised IPPO (de Witt et al., 2020). First, BC policies are trained on a large-scale dataset of human gameplay - 101,096 two-player games and 46,525 three-player games. Because BC alone struggles to generalise to unseen game states (Carroll et al., 2019; Hu et al., 2022; Bakhtin et al., 2022; Cornelisse & Vinitsky, 2024), we then refine them through regularised SP using IPPO. The regularisation ensures the final policies remain close to human play styles. The four proxy agents use varied architectures, hyperparameters and seeds, to capture a range of human playing strengths while maintaining human-like behaviour. We provide further details regarding PPO and IPPO in Appendix A.1.

When training human proxy agents using HDR-IPPO, we learn a parameterised policy, denoted as $\pi_\theta^{HP}$. As a first step, we train a BC policy, $\pi_\theta^{BC}$. Given Hanabi's discrete action space, this translates into a classification task. To capture the sequential nature of the observations, we adopt an LSTM-based architecture (Hochreiter & Schmidhuber, 1997). Our model takes a single player's trajectory as input and predicts an action for each timestep, conditioned on the AOH. Hence, for each player $i$, we maintain a memory component, $\phi_t^i$, which represents their observation history, $\tau_t^i$. The hidden state is updated for the next timestep using a function $h_{\theta^i}$, $\phi_{t+1}^i = h_{\theta^i}(o_t^i, \phi_t^i)$.

We train the BC model by minimising the standard cross-entropy loss between the predicted action distribution and the ground truth human actions. At each timestep, the agent acts greedily, selecting the action with the highest probability according to the policy, $a_t^i = \arg\max_a \pi_{\theta^i}^{BC}(a|o_t^i, \phi_t^i)$. At the end of each training epoch, we evaluate the BC policy in SP evaluation, storing the parameters, $\theta'$,

that yield the highest average score over a sample of games. In the second step of the HDR-IPPO method we leverage the baseline BC policy, $\pi_{\theta'}^{BC}$, to guide the training of a more robust policy, $\pi_{\theta}^{HP}$. To encourage the final policy to remain close to the human-like behaviour exhibited by $\pi_{\theta'}^{BC}$, we introduce a regularisation term based on the Kullback-Leibler (KL) (Kullback & Leibler, 1951) divergence between the action probability distributions of the two policies, a similar approach to what was explored in previous works (Cornelisse & Vinitsky, 2024; Hu et al., 2022; Bakhtin et al., 2022). This leads to the following KL divergence term for the policies at timestep $t$ for player $i$,

$$D_{\text{KL}}\left(\pi_{\theta'}^{BC}(\cdot|o_t^i, \phi_t^i)||\pi_{\theta}^{HP}(\cdot|o_t^i, \phi_t^i)\right) = \sum_{a \in \mathcal{A}} \pi_{\theta'}^{BC}(a|o_t^i, \phi_t^i) \log\left(\frac{\pi_{\theta'}^{BC}(a|o_t^i, \phi_t^i)}{\pi_{\theta}^{HP}(a|o_t^i, \phi_t^i)}\right). \quad (1)$$

This KL divergence term is then added to the standard IPPO (de Witt et al., 2020) objective, $\mathcal{L}_t^{\text{IPPO}}(\theta)$, with a regularisation weight $\lambda$ to control the strength of the regularisation. Therefore, we arrive at the following objective,

$$\mathcal{L}_t^{\text{HDR-IPPO}} = (1 - \lambda) \cdot \mathcal{L}_t^{\text{IPPO}} + \lambda \cdot D_{\text{KL}}\left(\pi_{\theta'}^{BC}(\cdot|o_t^i, \phi_t^i)||\pi_{\theta}^{HP}(\cdot|o_t^i, \phi_t^i)\right). \quad (2)$$

Further details and a complete list of hyperparameters used for training human proxy agents are available in Appendix A.4.

While this approach builds on prior research (Bakhtin et al., 2022; Hu et al., 2022; Cornelisse & Vinitsky, 2024), we design the following experiments to empirically validate that the generated human proxy agents exhibit human-like behaviour in Hanabi:

- **Cross-Play Analysis with BC Policies:** We test how well the human proxy agents coordinate with baseline BC policies, which closely follow human conventions but may lack generalisation. Strong performance in this cross-play setting confirms that our HDR-IPPO agents maintain human-compatible conventions while improving upon the generalisation limitations of pure BC.

- **Cross-Play Evaluation:** A common approach in the ZSC literature to demonstrate that agents converge to similar conventions is to pair them in cross-play evaluations. Hence, we assess cross-play performance between different human proxy agents trained with varying architectures, hyperparameters, and random seeds. Consistent high scores across these diverse pairings indicate convergence towards shared, human-like conventions.

- **Human Proxy Performance on Validation and Test Datasets**: We compare the performance of the BC policies on the validation and test datasets before HDR-IPPO training procedure with that of the final human proxies at the end of HDR-IPPO procedure. This comparison allows us to evaluate whether the agent retains human-like conventions present in the dataset.

Additionally, we present an ablation to study the impact of the HDR-IPPO KL regularisation term in the Appendix A.7. This analysis explores the effects of varying regularisation strength on the learned policies, offering further insights into the role of this component. We defer this discussion to the Appendix to maintain focus on the properties of the developed human proxy agents within the main text.

ACTION PREDICTION CHALLENGE

In addition to the primary human-AI ad-hoc coordination challenge, we introduce an action prediction challenge. While human-compatible play does not necessarily imply strong action prediction capabilities, accurate prediction of human actions provides further evidence of human-like behaviour. This challenge serves as a complementary evaluation metric, assessing how well an agent can anticipate human decisions in Hanabi. The action prediction dataset consists of a held-out portion of the human gameplay data used to train the human proxies. This data is distinct from both the training and validation sets used in the data-limited sub-challenges and is not open-sourced.

We evaluate performance using the teacher-forced cross-entropy loss since we want to quantify the difference between the predicted action distribution and the true human actions. Lower cross-entropy loss indicates better alignment with human decision-making. Participating agents receive a sequence of observations representing a game trajectory and must predict the action taken by the human player at each timestep.

EVALUATION API AND LEADERBOARD

To facilitate participation and ensure fair evaluation, we host the human proxy agents and provide access through a dedicated evaluation API. To initiate the evaluation process, participants must fill out a form to register for access to evaluation phase. Once registered, we provide participants with a private key, which grants restricted access to our human proxies (and test dataset). This key allows a one-time evaluation run, strictly limited to 1,000 games. Upon completion, results are published on the challenge leaderboard. Furthermore, our evaluation API allows interaction with the proxy agents while restricting access to global game state information, enforcing the partial observability inherent to Hanabi.

For the ad-hoc coordination challenge, the API provides access to each agent's local observations, mimicking the information available during real gameplay. Our API is inspired by Rutherford et al. (2023). We show an example for a single game evaluation in Snippet 2. For the action prediction challenge, the API provides batches of game trajectories from the held-out human gameplay dataset. Further details and example usage of the action prediction API are available in our repository.

```python
# Connect to evaluation server.
eval_space = EvaluationSpace(submission_key="key")
# Create candidate agent.
candidate_agent = MyAgent(...)
# Get first environment for evaluation.
env = eval_space.next_environment()
obs, legal_moves = env.reset()
done, game_score = False, 0
while not done:
    # Step environment based on local observation.
    action = candidate_agent.act(obs, legal_moves)
    obs, reward, done, legal_moves = env.step(action)
    game_score += reward
# Once done, we can get next environment.
env = eval_space.next_environment()
```

Figure 2: Ad-hoc teamplay evaluation API. Through an API we give limited access to human proxy agents. With a single API key, candidates get access to a limited number of games for evaluation.

## 5 EXPERIMENTS AND RESULTS

### 5.1 VALIDATING HUMAN PROXIES

Table 2 presents the SP performance of our final human proxy agents and their improvement over the initial BC policies. The performance gains from regularised RL are particularly pronounced in the three-player setting, where the BC policies trained on limited data frequently lose all their lives, resulting in a large proportion of zero-score games and overall bad performance. For instance, in a three-player SP evaluation, BC agents scored zero in 70.92% of games. With the help of regularised RL, human proxies score zero points only on 0.27% of the games. This highlights the robustness achieved through regularised SP RL. Moreover, we observe a larger number of perfect-score games for all agents, compared to BC counterparts. Next, we provide evidence that our agents produce human-like strategies, aligning with those observed in our dataset.

**Cross-Play Analysis with BC Policies.** We know that BC policies closely adhere to human conventions but lack generalisation capabilities. Therefore, we would expect the cross-play scores with HDR-IPPO to be high (and even higher than BC SP). Also, median scores should be higher than mean scores, given that we expect to see a lot of games with scores of zero, where BC agents fail when out of distribution. We observe in Figures 3a and 3b that these agents coordinate well, as expected. This suggests that HDR-IPPO agents have not only learned to play effectively but have also retained strategies employed by agents trained solely on human data. Notably, median scores are very similar in all the combinations of agents, showing evidence of successful coordination. In

Table 2: SP evaluation results for human proxies. We compare human proxy results with the results of respective BC policies. For SP evaluation, we report mean $\pm$ SE over 5,000 games.

| Metric | $\pi_{\theta 1}^{HP}$ | $\pi_{\theta 2}^{HP}$ | $\pi_{\theta 3}^{HP}$ | $\pi_{\theta 4}^{HP}$ |
|---|---|---|---|---|
| Mean Self-Play Score | $22.55 \pm 0.03$ | $22.97 \pm 0.03$ | $20.88 \pm 0.03$ | $21.21 \pm 0.03$ |
| Improvement over BC | 3.0 | 4.0 | 15.7 | 13.9 |
| Perfect Games (%) | 23.86% | 29.66% | 2.76% | 3.88% |
| Perfect Games BC (%) | 16.12% | 19.88% | 1.34% | 1.80% |
| Zero-Score Games (%) | 0.10% | 0.04% | 0.34% | 0.20% |
| Zero-Score Games BC (%) | 11.42% | 17.70% | 75.82% | 66.02% |

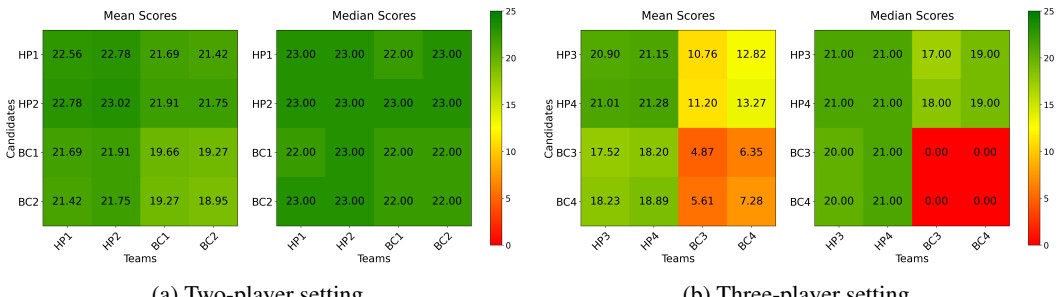

(a) Two-player setting.      (b) Three-player setting.

Figure 3: Cross-play performance matrices for human proxy agents and corresponding BC policies in two-player and three-player settings. We show strong performance across different pairings indicating that policies converged to similar conventions. Each element $(i, j)$ represents the average score achieved by different team configurations, averaged over all possible permutations of player positions. For the three-player setting team comprises two instances of agent $j$ and one instance of agent $i$. The average score for each team is calculated based on 15,000 games per permutation.

the three-player setting, we notice a substantial difference between the median and mean scores, particularly when two BC policies are paired with a single human proxy. This indicates a higher frequency of zero-score games, which aligns with the known brittleness of BC policies in unfamiliar situations. However, the scores are still high, even though the BC policies achieve poor performance in SP. Even when pairing two BC policies with a single stronger human proxy, the score increases significantly resulting in the median score of 17 or more in all configurations. These cross-play experiments offer evidence that human proxy agents have successfully learned to play the game at a high level while retaining the ability to interact effectively with agents employing strategies derived purely from human demonstrations.

**Cross-Play Analysis for Human Proxies.** Furthermore, Figure 3a and 3b illustrate the cross-play results for two-player and three-player setting human proxy agents. We observe consistent scores across different pairings, suggesting that the agents have converged to compatible strategies despite variations in their architectures and regularisation strengths. Interestingly, pairing a weaker agent with a stronger one results in scores surpassing the weaker agent's SP performance. This further reinforces the notion that the agents have learned similar conventions, enabling them to coordinate effectively even when paired with partners of differing skill levels.

**Human Proxies Performance on Test and Validation Datasets.** Next, we evaluate human proxies on the validation and test set that was held out for the training of the BC agents. The results are reported in Table 3. Here, we observe similar performance in terms of accuracy and loss on the test and validation sets as for the BC policy.

We refer the reader to Appendix A.4 and A.7 for additional experiments and results. We design an ablation study where we vary the strength of the regularisation term and examine the effects on the final policy. Additionally, we show that starting the training from a BC policy, without regularisation, yields weak and/or human-incompatible policies.

Table 3: Loss and accuracy of human proxy policies on the hold-out validation and test sets. Compared to respective BC policies, human proxies significantly improve in SP, while still fitting the validation and test sets well.

| Metric | $\pi_{\theta 1}^{HP}$ | $\pi_{\theta 2}^{HP}$ | $\pi_{\theta 3}^{HP}$ | $\pi_{\theta 4}^{HP}$ |
|---|---|---|---|---|
| Number of Players | 2 | 2 | 3 | 3 |
| Accuracy | 0.63 | 0.63 | 0.43 | 0.44 |
| Accuracy Difference to BC | -0.03 | -0.08 | -0.08 | -0.07 |
| Loss | 0.53 | 0.54 | 0.63 | 0.60 |
| Loss Difference to BC | +0.05 | +0.07 | +0.08 | +0.06 |

## 5.2 BASELINES

We consider a variety of baseline methods, informed by previous research (Hu et al., 2021; Carroll et al., 2019). **IPPO**: Policy trained in SP, without human data. **BC**: Policy trained only on available human data. Acts as a reference policy. **HDR-IPPO**: Policy builds upon the BC reference by incorporating regularised RL. **BR-BC** (Carroll et al., 2019): Policy is trained by embedding the BC policy in the environment, treating its actions as a part of environment dynamics during SP learning. **OBL** (Hu et al., 2021): Strong approach for ZSC that doesn't utilise any human data during training. We use this agent only in the two-player setting since we do not have access to three-player weights. **Other-Play (OP)** (Hu et al., 2020): A ZSC method that prevents agents from learning equivalent but mutually incompatible policies across independent training runs. OP accomplishes this by enforcing the equivariance of the policies under the symmetries of the Dec-POMDP, which must be provided as an input of the algorithm.

We adopt the IPPO configuration from Rutherford et al. (2023) and adapt the BR-BC training procedure from Carroll et al. (2019). These baseline approaches utilise feed-forward architectures. Our BC and HDR-IPPO baselines, in contrast, employ an LSTM-based architecture. While based on the same backbone as the human proxy agents, we use different hyperparameters optimised for the data-limited challenge settings. Additionally, we utilise different random seeds than those used for training the human proxies.

For each of the BC, BR-BC, and HDR-IPPO baselines, we train with three different random seeds. The best-performing agent, based on cross-entropy loss on the validation set, is selected for evaluation. Notably, we observe minimal performance variance across different seeds on the validation set (see Appendix A.2). The IPPO baseline, intended to showcase the limitations of SP in ad-hoc coordination, is trained with a single seed. Finally, we utilise pre-trained weights for the OBL baseline and respective hyperparameters (Hu et al., 2021).

Table 4 presents the initial AH2AC2 leaderboard. OBL (L4) achieves the highest performance, despite not using any human data at all. BR-BC performs significantly better than HDR-IPPO in the two-player setting. In the three-player setting, HDR-IPPO achieves the best results and the gap compared to human-proxy SP is considerably larger. Unfortunately, there are no pre-trained weights for OBL available for this setting. The BC and IPPO baselines perform poorly in both the two- and three-player settings, as expected. Our findings validate the strength of OBL, which outperforms BC, BR-BC, and HDR-IPPO, despite not utilizing any human data. OP, which aims to break symmetries to avoid arbitrary conventions, is insufficient for successful human-AI coordination. Furthermore, our results reveal that current methods designed to leverage limited human data as an inductive bias underperform compared to SOTA ZSC algorithms like OBL. This indicates a gap in the current research landscape: existing approaches like BR-BC and HDR-IPPO are inadequate for effectively integrating very small datasets from human populations to enhance coordination. Consequently, there is a pressing need for new methods capable of efficiently utilizing limited human data to guide learning.

Table 4: AH2AC2 leaderboard. OBL does not make use of available human data, yet it achieves high scores. Methods that use a combination of supervised learning and RL generally outperform methods that use only supervised learning or only RL.

| Players | Method | Mean Scores | Median Scores | Cross-Entropy |
|---|---|---|---|---|
| | OBL (L4) | **21.04** | 22 | 1.33 |
| | BR-BC | 19.41 | 20 | 10.82 |
| **2P** | OP | 13.91 | 19 | 7.81 |
| | HDR-IPPO | 12.76 | 15 | 0.96 |
| | IPPO | 10.16 | 14 | 12.60 |
| | BC | 2.12 | 0 | 0.86 |
| | *Human Proxies* [†] | 22.76 | 23 | 0.54 |
| | *BR-BC*[*†] | 22.59 | 23 | 5.00 |
| | HDR-IPPO | **14.03** | 16 | 0.80 |
| | OP | 12.87 | 18 | 6.40 |
| **3P** | BR-BC | 11.89 | 12 | 29.89 |
| | IPPO | 6.34 | 0 | 8.60 |
| | BC | 3.31 | 0 | 0.70 |
| | *Human Proxies* [†] | 20.86 | 21 | 0.62 |
| | *BR-BC*[*†] | 18.80 | 19 | 7.53 |

[†] Not constrained by game limits, acts as a golden standard. BR-BC* is trained with a BC policy trained on the entire dataset. We report average performance over two human proxies.

# 6    CONCLUSION

We introduced the Ad-Hoc Human-AI Coordination Challenge (AH2AC2), evaluating human-AI ad-hoc team play in the context of the cooperative card game Hanabi. By leveraging a large-scale human play dataset to generate human-like agent policies, we provide a meaningful evaluation framework for assessing AI agents' ability to coordinate with human partners. We released a comprehensive set of baselines, including agents trained with and without human data, to provide a reference point for evaluating novel approaches. Our results highlight the inherent challenge of human-AI coordination. We believe that AH2AC2 represents a significant step forward in the field of human-AI coordination. By providing a standardised evaluation protocol, human proxy agents, and a diverse set of baselines, we aim to foster further research and development. We invite researchers and practitioners alike to participate, pushing the boundaries of what's possible in human-AI collaboration.

Several open challenges and questions remain. We highlight some promising directions for future research:

- **Theoretical Analysis of HDR-IPPO:** While our experiments and previous works provide strong empirical evidence for the effectiveness of regularised RL in generating human-like agents, a deeper theoretical understanding of the methodology is crucial.

- **Generalisation of the benchmark** We currently only cover 2 and 3 players as well as "standard" Hanabi, ignoring the large set of possible variations of the game that are created e.g. by the "rainbow cards". Extending the benchmark to cover those scenarios would be a great way to measure the generalisation ability of agentic systems.

- **Direct Human-AI Play with Human Proxy Agents:** The ultimate validation of our human proxy agents requires direct human-AI play. Future work should involve conducting play experiments with human participants, comparing their experiences and performance when playing with human proxies versus playing with other humans.

- **Extension to Agentic Large Language Models (LLMs):** The conventions used in our dataset are documented using human language. This opens up the possibility of extending our benchmark to include agentic LLMs as baselines. Excitingly, in combination with our human proxies Hanabi turns into a fantastic benchmark for the theory of mind and human coordination of agentic LLMs in complex partially observable settings.

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

## A  APPENDIX

### A.1  FURTHER BACKGROUND

#### INDEPENDENT PPO

Proximal Policy Optimisation (PPO) Schulman et al. (2017) is a method initially developed for single-agent RL. PPO aims to address performance collapse in policy gradient methods. It does this by bounding the ratio of action probabilities between the old and new policies. PPO optimises the objective function,

$$\mathbb{E}_{s \sim d^\pi, a \sim \pi} \left[ \min \left( \frac{\tilde{\pi}(a \mid s)}{\pi(a \mid s)} A^\pi(s,a), \text{clip} \left( \frac{\tilde{\pi}(a \mid s)}{\pi(a \mid s)}, 1 - \epsilon, 1 + \epsilon \right) A^\pi(s,a) \right) \right], \qquad (3)$$

where $\text{clip}(t, a, b)$ is a function that outputs $a$ if $t < a, b$ if $t > b$ and $t$ otherwise. We consider the extension to multi-agent setting by using independent learning (de Witt et al., 2020). Here, each agent treats the others as part of the environment and learns a critic using its local AOH.

#### OBL: ZERO-SHOT COORDINATION

Most SP methods exhibit a tendency towards brittleness and over-coordination, leading to suboptimal performance when paired with independently trained policies. To address this, OBL (Hu et al., 2021) was proposed as an algorithm that learns optimal grounded policies without relying on arbitrary conventions or assumptions about other agents. OBL has demonstrated remarkable success in ZSC scenarios, making it a compelling baseline for ad-hoc teamwork, even though it does not utilise any of the human data provided for the challenge.

### A.2  ADDITIONAL RESULTS

We provide additional results for performance on validation and test sets in Table 5 and 6.

Table 5: Cross-entropy loss on the validation and test sets for BC, BR-BC and HDR-IPPO agents in the two-player setting. For human proxies we report results over two available agents. For the rest, we report results over three different seeds. Even though agents are trained with different seeds, they achieve almost identical results. We report loss $\pm SE$.

| Data Used | Method | Validation Set | | Test Set | |
|---|---|---|---|---|---|
| | | Loss | Accuracy | Loss | Accuracy |
| **1,000 Games** | BC | $0.87 \pm 0.0$ | $0.46 \pm 0.0$ | $0.87 \pm 0.0$ | $0.46 \pm 0.0$ |
| | BR-BC | $10.93 \pm 0.1$ | $0.25 \pm 0.0$ | $10.96 \pm 0.1$ | $0.25 \pm 0.0$ |
| | HDR-IPPO | $0.96 \pm 0.0$ | $0.41 \pm 0.0$ | $0.97 \pm 0.0$ | $0.40 \pm 0.0$ |
| **Entire Dataset** | BC | $0.47 \pm 0.0$ | $0.67 \pm 0.0$ | $0.48 \pm 0.0$ | $0.67 \pm 0.0$ |
| | Human Proxy | $0.53 \pm 0.0$ | $0.63 \pm 0.0$ | $0.54 \pm 0.0$ | $0.63 \pm 0.0$ |

Table 6: Cross-entropy loss on the validation and test sets for BC, BR-BC and HDR-IPPO agents in the three-player setting. For human proxies we report results over two available agents. For the rest, we report results over three different seeds. Even though agents are trained with different seeds, they achieve almost identical results. We report loss $\pm SE$.

| Data Used | Method | Validation Set | | Test Set | |
|---|---|---|---|---|---|
| | | Loss | Accuracy | Loss | Accuracy |
| **1,000 Games** | BC | $0.71 \pm 0.0$ | $0.39 \pm 0.0$ | $0.70 \pm 0.0$ | $0.40 \pm 0.0$ |
| | BR-BC | $32.30 \pm 1.4$ | $0.07 \pm 0.0$ | $32.48 \pm 1.6$ | $0.07 \pm 0.0$ |
| | HDR-IPPO | $0.81 \pm 0.0$ | $0.31 \pm 0.0$ | $0.81 \pm 0.0$ | $0.31 \pm 0.0$ |
| **Entire Dataset** | BC | $0.55 \pm 0.0$ | $0.51 \pm 0.0$ | $0.54 \pm 0.0$ | $0.51 \pm 0.0$ |
| | Human Proxy | $0.62 \pm 0.0$ | $0.43 \pm 0.0$ | $0.62 \pm 0.0$ | $0.44 \pm 0.0$ |

### A.3 TRAINING DETAILS

#### BEHAVIOURAL CLONING (BC)

For each sampled mini-batch of games, we extract individual player trajectories, effectively decomposing each game into $n$ trajectories, where $n$ represents the number of players. This yields sets of trajectories, denoted by $U_d = \{u_0, ..., u_n\}$, with one trajectory per player for each game. Each trajectory, $u_i = \{(o_t^i, a_t^i)\}_{t=1}^T$, consists of local player observations, $o_t^i \in \mathcal{O}^i$, and corresponding actions, $a_t^i \in \mathcal{A}^i$, taken at timestep $t$. Our BC model takes a single player's trajectory as input and predicts an action for each timestep, conditioned on the AOH. A mini-batch of size $b$ is constructed by concatenating multiple sets of trajectories: $\mathcal{D}_{\text{batch}} = U_0 \| ... \| U_b$, where $\|$ denotes concatenation. To augment the training data and improve generalisation, we randomly shuffle the colour space for both observations and actions within each mini-batch before feeding it to the network, as done in the previous works to enhance performance of our model (Hu et al., 2021).

The loss for each batch is calculated as

$$\mathcal{L}^{BC}(\theta) = -\frac{1}{\sum_{\tau_i \in \mathcal{D}_{batch}} |\tau_i|} \sum_{\tau_i \in \mathcal{D}_{batch}} \sum_{(o_t, a_t) \in \tau_i} \ell(\pi_\theta^{BC}(a|o_t, \phi_t), a_t), \quad (4)$$

where $\mathcal{D}_{batch}$ is the batch of trajectories $\tau_i$ and $\ell$ is the standard cross-entropy loss. The policy, $\pi_\theta^{BC}$, is conditioned on both the hidden state $\phi_t^i$, encoding the observation history, and the current local observation $o_t^i$.

#### BEST RESPONSE TO BEHAVIOURAL CLONING (BR-BC)

When training with an embedded human model, we initially train in SP and anneal the amount of SP linearly to zero. Then, we continue training with the BC policy, as Carroll et al. (2019) find that this improves agents' performance. Additionally, we consider the three agent setting that was not considered by Carroll et al. (2019). When annealing, based on the annealing factor, we sample agents to decide whether we train in SP, with a single BC policy or with two BC policies.

### A.4 HUMAN PROXIES: ADDITIONAL DETAILS

#### DATASET

Dataset used for training human proxies contains 147,621 Hanabi games, comprising 101,096 two-player and 46,525 three-player games. Table 7 summarises key statistics for scores and game lengths across both player configurations.

Table 7: The dataset used for training human proxies comprises a total of 147,621 games, including 101,096 two-player and 46,525 three-player games. The table presents descriptive statistics for game scores and lengths across both player configurations.

| Setting | Metric | Min | Max | Avg | Median | Std |
|---|---|---|---|---|---|---|
| Two-Player | Scores | 1 | 25 | 23.09 | 24.00 | 2.16 |
| | Game Lengths | 2 | 88 | 65.70 | 66.00 | 3.63 |
| Three-Player | Scores | 2 | 25 | 22.94 | 23.00 | 2.10 |
| | Game Lengths | 34 | 78 | 58.38 | 59.00 | 3.36 |

#### ARCHITECTURES

The architectures and hyperparameters used to train human proxy agents are shown in Table 8. Each model includes a fully connected layer, a multi-layer LSTM block, and a decoder fully connected network that maps the LSTM encodings to a probability distribution over actions. In our implementation, we employ loss masking to ensure that the probability of selecting an illegal action is effectively set to zero during sampling from the policy. Optimisation is performed using the Adam optimiser (Kingma & Ba, 2014) across all configurations. When applying a linear learning rate schedule, the learning rate is reduced to its minimum allowable value during the final 10% of the training process.

#### HYPERPARAMETER SEARCH

We run a hyperparameter search for the BC policies used for training human proxies. We performed a full grid search across a range of hyperparameters, considering both two-player and three-player game settings. The hyperparameter search space is shown in Table 9.

From these configurations, we selected the two best-performing settings for both two-player and three-player scenarios. These configurations were subsequently employed in the second step of the HDR-IPPO procedure. Respective agents are what we refer to as human proxies.

#### ROLE OF REGULARISATION

Our human proxy policies have demonstrated successful coordination in cross-play with both other human proxies and the original BC policies. Here, we show that arbitrary policies fail to coordinate well with them. We introduce a new set of agents trained using the same HDR-IPPO procedure as our human proxies, but with a crucial difference, the human data regularisation weight is set to zero, $\lambda = 0$.

These new policies start from the same BC policy weights as our human proxies and utilise identical hyperparameters, except for the KL regularisation term, which is eliminated. We focus on two specific agents, one for the two-player setting (based on $\pi_{\theta^2}^{BC}$) and one for the three-player setting (based on $\pi_{\theta^4}^{BC}$). We use the weights obtained at the final training timestep as checkpoints for these model weights.

Firstly, we observed a rapid deterioration in SP performance for the non-regularised policies. The agents quickly lost the ability to perform well, even in SP. This degradation is evident in Figure 4 and Figure 5. This sharp decline suggests a divergence from the strategies initially learned during the BC phase. Without the regularisation term guiding the policy towards human-like behaviour, the agents appear to explore alternative strategies that may lead to suboptimal performance in the context of the game.

Table 8: Human proxy agent training configurations and architectures. We showcase both BC and IPPO hyperparameters in a single table.

| Hyperparameter | $\pi_{\theta 1}^{HP}$ | $\pi_{\theta 2}^{HP}$ | $\pi_{\theta 3}^{HP}$ | $\pi_{\theta 4}^{HP}$ |
|---|---|---|---|---|
| **Network Architecture** | | | | |
| Num Players | 2 | 2 | 3 | 3 |
| Activation | GELU | GELU | GELU | GELU |
| LSTM Layers | 512, 512, 512 | 512, 512 | 512, 512, 512 | 512, 512 |
| Input Embedding | 1024 | 1024 | 1024 | 1024 |
| Decoder MLP | 512 | 256 | 1024 | 1024 |
| **BC** | | | | |
| **Optimisation** | | | | |
| Batch Size | 256 | 256 | 128 | 256 |
| Dropout | 0.5 | 0.3 | 0.5 | 0.5 |
| LR Schedule | Linear | Linear | Linear | Linear |
| Initial LR | 0.005 | 0.005 | 0.005 | 0.005 |
| Final LR | 5.0e-05 | 5.0e-05 | 1.0e-05 | 1.0e-05 |
| Epochs | 50 | 50 | 70 | 70 |
| **Training** | | | | |
| Permute Colours | Yes | Yes | Yes | Yes |
| Self-Play Eval Games | 5000 | 5000 | 5000 | 5000 |
| **IPPO** | | | | |
| **Optimisation** | | | | |
| Learning Rate | 0.0003 | 0.0003 | 0.0005 | 0.0003 |
| Gamma Discount | 0.99 | 0.99 | 0.99 | 0.99 |
| GAE Lambda | 0.95 | 0.95 | 0.95 | 0.95 |
| Clip Epsilon | 0.2 | 0.2 | 0.2 | 0.2 |
| Entropy Coefficient | 1.0e-05 | 0.0001 | 0.0001 | 0.0001 |
| Value Function Coeff | 0.5 | 0.5 | 0.5 | 0.5 |
| Max Gradient Norm | 0.5 | 0.5 | 0.5 | 0.5 |
| Update Epochs | 4 | 4 | 4 | 4 |
| Num Minibatches | 4 | 4 | 4 | 4 |
| **Critic Network** | | | | |
| Critic MLP | 512 | 512 | 512 | 512 |
| **Training** | | | | |
| BC Policy KL Weight | 0.3 | 0.2 | 0.1 | 0.15 |
| Total Timesteps | 5e9 | 5e9 | 5e9 | 5e9 |
| **Environments** | | | | |
| Num Env Steps | 90 | 90 | 90 | 90 |
| Num Train Envs | 1024 | 1024 | 1024 | 1024 |
| Num Eval Envs | 512 | 512 | 512 | 512 |

Table 9: Hyperparameter search space for BC when developing human proxies.

| Hyperparameter | Values |
|---|---|
| Dropout | 0.3, 0.5 |
| Batch Size | 64, 128, 256 |
| LSTM Layers | (512, 512), (512, 512, 512), (1024, 1024), (1024, 1024, 1024) |
| Input Embedding | 512, 1024, (512, 512) |
| Decoder MLP | 512, 1024 |

After the initial performance drop, the scores of the non-regularised policies gradually improve during training. However, with the current set of hyperparameters, this progress is slow and we do not observe convergence. The agents seem unable to efficiently re-learn a different, yet still effective, policy. It is important to note that this behaviour is not observed with different hyperparameter settings, but for the purpose of direct comparison in this analysis, we maintained the same configuration as the human proxy agents.

When paired with human proxies and BC policies, the non-regularised agents exhibit significantly lower coordination and overall performance compared to the pairings between human proxies and

BC policies exclusively. This discrepancy highlights the crucial role of the human data regularisation term in ensuring that the learned strategies remain aligned with human-like play, facilitating successful coordination.

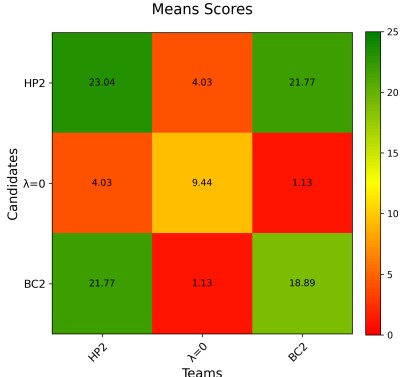 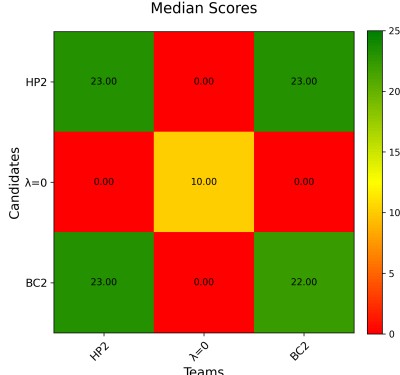

Figure 4: Cross-play performance matrix for two-player agents, comparing a human proxy agent ($\pi_{\theta^2}^{HP}$), its corresponding BC policy ($\pi_{\theta^2}^{BC}$), and a non-regularised HDR-IPPO agent initialised from the same BC policy, $\pi_{\theta^2}^{BC}$, but where $\lambda = 0$. Each matrix element represents the average score achieved by a team, averaged over both player orderings. Averages are calculated based on 15,000 games per team permutation.

Consider the three-player cross-play matrix depicted in Figure 5. The BC policy, while achieving a low SP score of 7.19, significantly improves to 18.92 when paired with two human proxies. This substantial boost underscores the BC policy's inherent understanding of human-like strategies, even if it struggles to execute them independently. In contrast, the non-regularised policy, with a SP score of 3.99, only sees a marginal improvement to 6.93 in cross-play with human proxies. Furthermore, pairing the BC policy (the stronger player in SP) with two non-regularised policies actually decreases the performance compared to the SP performance of non-regularised policies. This degradation suggests a fundamental incompatibility between the strategies learned by the non-regularised policy and the human-like conventions given by the BC policy.

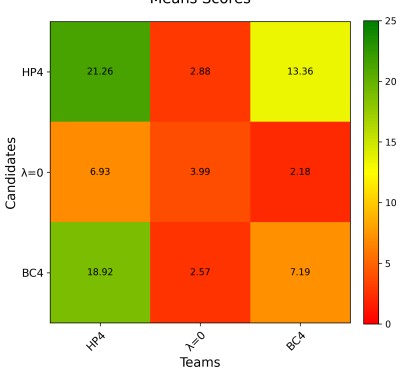 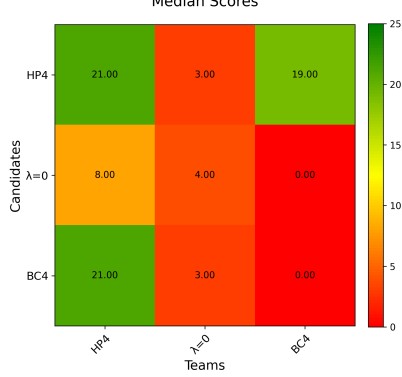

Figure 5: Cross-play performance matrix for three-player setting, with scores for a human proxy agent ($\pi_{\theta^4}^{HP}$), its corresponding BC policy ($\pi_{\theta^4}^{BC}$), and a non-regularised HDR-IPPO agent initialised from the same BC policy, $\pi_{\theta^4}^{BC}$, but with $\lambda = 0$. Each matrix element $(i, j)$ represents the average score achieved by a team composed of two instances of one agent on the $x$-axis and one instance of agent on the $y$-axis, averaged over all possible permutations of player positions. Averages are calculated based on 15,000 games per team permutation.

### A.5  Behaviour Analysis

The dataset we use in this work comes from the human policies that adhere to H-Group Conventions. This distinguishes our work from previous human-AI coordination studies, as these strategies can be explicitly described in natural language and are documented.

We also include behavioural metrics proposed by (Canaan et al., 2020), for both the trajectories found in the dataset and the ones created by the human proxies. Results are shown in Table 10.

Table 10: We compare behaviour features available in the dataset and once present in trajectories generated by our human proxies. Precisely, we compute Information per Play (IPP) and Communicativeness. We show metrics are almost identical for both cases.

| Trajectories | Proxy | IPP | Communicativeness |
|---|---|---|---|
| Two-Player | 2P Dataset | 0.44 | 0.47 |
| | $\pi_{\theta^1}^{HP}$ | 0.43 | 0.45 |
| | $\pi_{\theta^2}^{HP}$ | 0.44 | 0.48 |
| Three-Player | 3P Dataset | 0.42 | 0.49 |
| | $\pi_{\theta^3}^{HP}$ | 0.44 | 0.47 |
| | $\pi_{\theta^4}^{HP}$ | 0.44 | 0.46 |

Furthermore, we provide a qualitative assessment by analysing games played by our human proxies. Our findings show that our agents exhibit highly human-like behaviour and in most instances adhere to H-convention strategies. Sample renders can be found in the anonymous repository we provide and here we analyse the first game provided. We analyse each of the moves below, marking each move as a success or failure, depending on whether the convention is followed correctly. We also find that the mistakes made when not following conventions are very human-like mistakes. For example, on Turn 2, the agent violates the good touch principle, but this move also looks a lot like a 2 save. Hence, it violates a convention because it tried to follow a different one, which was not applicable in this case. Most of the failure cases we found come from confusing conventions that should be played in that particular instance, which is a human-like mistake.

1. Turn 0: Success (Hint blue, play clue on the blue 1)
2. Turn 1: Success (Play clue on both the 1's)
3. Turn 2: Failure (Looks like a 2 save, but doubles the yellow 2's, violating good touch principle)
4. Turn 3: Success (Actor 1 plays card slot 2, knowing it's a B1, which follows the "play clue" convention)
5. Turn 4: Success (Actor 0 gives a 5 save when Red 5 is on the chop)
6. Turn 5: Success (Actor 1 plays a known playable card)
7. Turn 6: Failure (Actor 0 should have 2-saved the Red 2 on chop)
8. Turn 7: Success (Actor 1 discards Red 2, the chop card)
9. Turn 8: Success (Actor 0 plays slot 0, a known playable 1)
10. Turn 9: Success (Actor 1 discards chop on slot 3)
11. Turn 10: Success (Fix clued the duplicate cards)
12. Turn 11: Success (Plays known playable card)
13. Turn 12: Success (Gives play clue to the Red 1)
14. Turn 13: Success (Plays the Red 1)
15. Turn 14: Success (Discards chop)

16. Turn 15: Success (Gives play clue to the Yellow 3)

17. Turn 16: Success (Plays Yellow 3)

18. Turn 17: Success (Discards known trash)

19. Turn 18: Success (Discards chop)

20. Turn 19: Failure (Doesn't understand chop-focus, giving play clue to wrong card, and violates good touch principle by clueing Red 1 and Red 3 twice)

21. Turn 20: Failure (The focus of the last clue was the chop, so it should have played the Red 3, but played Red 2 instead)

22. Turn 21: Success (Fix clue on the duplicated Red 3's)

23. Turn 22: Success (Plays known playable Red 3)

24. Turn 23: Success (Discards chop, position 1)

25. Turn 24: Success (Gives play clue to White 2)

26. Turn 25: Success (Plays White 2)

27. Turn 26: Success (Play clue on the Red 4)

28. Turn 27: Success (Plays the Red 4)

29. Turn 28: Success (Play clue on Blue 4, and filling in White 3)

30. Turn 29: Success (Plays White 3)

31. Turn 30: Success (Discards known trash)

32. Turn 31: Success (Play clue on White 4)

33. Turn 32: Success (Plays White 4)

34. Turn 33: Success (Plays known playable Red 5)

35. Turn 34: Success (Discards known trash Red 1)

36. Turn 35: Failure (Should have played its Blue 3 because of the play clue, instead gave a 5 hint off chop, which is illegal after the late game)

37. Turn 36: Success (Discards chop)

38. Turn 37: Failure (Should have played its Blue 3, instead gave a 2 hint which is illegal)

39. Turn 38: Success (Discards chop)

40. Turn 39: Failure (Should have played its Blue 3, instead discarded chop)

41. Turn 40: Success (Play clue on Blue 4, filling in Blue 3)

42. Turn 41: Success (Plays Blue 3)

43. Turn 42: Success (Discards chop)

44. Turn 43: Success (Plays Blue 4)

45. Turn 44: Success (Discards chop)

46. Turn 45: Success (Hint Blue, filling in Blue 5)

47. Turn 46: Success (Plays Blue 5)

48. Turn 47: Success (Play clue on Green 1)

49. Turn 48: Success (Plays Green 1)

50. Turn 49: Success (Discards chop)

51. Turn 50: Success (Play clue on Yellow 4)

52. Turn 51: Success (Plays Yellow 4)

53. Turn 52: Success (Plays Green 2)

54. Turn 53: Success (Discards chop)

55. Turn 54: Success (Reveals Green 4 identity)

56. Turn 55: Success (Discards chop)

57. Turn 56: Success (Plays Yellow 5)

58. Turn 57: Success (Discards chop)

59. Turn 58: Success (5 save on Green 5)

60. Turn 59: Success (Stalling, hinting 1s)

61. Turn 60: Failure (Hinting Green is seen as a play clue on Green 1, which is illegal)

62. Turn 61: Success (Plays Green 1, which it thought was Green 3 because of convention)

63. Turn 62: Success (Play clue on Green 3)

64. Turn 63: Success (Plays Green 3)

65. Turn 64: Success (Hints White 5)

66. Turn 65: Success (Plays Green 4)

In summary, in this game, the human proxy followed H-group conventions for 88% of the moves and used various strategies while playing the game.

- **Successful H-Group conventions played:**
  - Giving play clue (14x)
  - Responding to play clue (23x)
  - 2 Save (1x)
  - 5 Save (2x)
  - Discard chop (13x)
  - Fix clue (3x)
  - Discard known trash (3x)
  - Stall clue (1x)

- **H-Group Conventions violated:**
  - Violating Good touch Principle (1x)
  - Failure to 2 save (1x)
  - Incorrectly gives chop focus clue (1x)
  - Incorrectly responding to chop focus clue (1x)
  - Failure to play play clue (3x)
  - Incorrectly give play clue (1x)

A.6    AH2AC2: CHALLENGE IMPLEMENTATION DETAILS

We have established a dedicated website for the AH2AC2 challenge at https://ah2ac2.com/. Participants can request to join the challenge through this website, where they will also find a leaderboard displaying existing results.

Upon submitting a participation request, candidates will be notified when the challenge becomes public and will receive an API key for testing their implementations and for official evaluation. The results of candidate agents will automatically appear on the leaderboard once all games are completed. Participation in the action prediction challenge is currently optional but encouraged.

For the evaluation, we assess the performance of each candidate agent over a total of 2,000 games played with our human proxy agents: 1,000 games in the two-player setting and 1,000 games in the three-player setting. We consider all seating configurations and all combinations of agents and seating positions, including scenarios where candidates control multiple agents (except for self-play situations). The evaluation is conducted uniformly across different seating configurations to ensure a fair and comprehensive assessment. This setup follows the procedure introduced in the seminal work on ad hoc teamwork evaluation by (Stone et al., 2010). Participants can obtain evaluation results and information for their agents at any time through our dedicated API.

## A.7 HDR-IPPO: ABLATION STUDY

We conduct an ablation study to investigate the impact of the human data regularisation term on the performance and behaviour of agents trained with the HDR-IPPO. By systematically varying the strength of the regularisation, we aim to gain insights into its role in guiding policy learning towards strategies learned during BC (in this case, human-like strategies).

### METHODOLOGY

We focus on the two-player setting, where BC has demonstrated strong performance as a baseline. Due to the computational and time constraints associated with training HDR-IPPO agents, it was not feasible to extend this study to the three-player setting. To systematically analyse the effect of the KL regularisation term introduced in the HDR-IPPO algorithm, we adopt the following methodology:

1. We train a new BC policy that serves as both a starting point for subsequent HDR-IPPO training and a baseline for comparison in this ablation study.

2. From the baseline BC policy, we train multiple HDR-IPPO agents. These agents share identical architectures, hyperparameters, and training procedures, with the sole exception of the human data regularisation weight, $\lambda$. We vary the weight of regularisation term across a range of values, from no regularisation ($\lambda = 0.00$) to very high regularisation ($\lambda = 0.70$) to comprehensively investigate the effects of this term during both training and evaluation. The final policy weights for each agent are stored for subsequent analysis.

3. We evaluate, analyse and compare the trained HDR-IPPO agents and the baseline BC policy. Precisely, we:

   (a) Assess the SP performance of each agent to understand how the strength of KL regularisation influences its ability to play Hanabi effectively on its own.

   (b) Evaluate the cross-play performance of each HDR-IPPO agent when paired with the baseline BC policy. Higher scores in this setting indicate that the HDR-IPPO agent has learned strategies compatible with the human-like conventions exhibited by the BC agent.

   (c) Conduct cross-play evaluations among different HDR-IPPO agents to identify potential coordination patterns.

   (d) Evaluate each agent on the held-out validation and test sets of human data. Good performance on this data indicates that the agent has effectively retained the conventions observed in the human demonstrations.

   (e) Plot the KL divergence between the action distributions of each HDR-IPPO agent and the baseline BC policy throughout the training process. This visualisation allows us to identify potential convergence or divergence patterns for the KL divergence term.

### EXPERIMENTAL SETUP

The hyperparameters used for training the agents in this ablation study are listed in Table 11.

Table 11: Hyperparameters used for training agents in the ablation study. We showcase both BC and IPPO hyperparameters in a single table.

| Hyperparameter | Value |
| --- | --- |
| **Network** | |
| Num Players | 2 |
| Activation | GELU |
| LSTM Layers | 512 |
| Input Embedding | 1024 |
| Decoder MLP | 512 |
| Dropout | 0.5 |
| **BC** | |
| **Optimisation** | |
| Optimiser | Adam (Kingma & Ba, 2014) |
| Batch Size | 128 |
| LR Schedule | Linear |
| Initial LR | 0.005 |
| Final LR | 1.0e-05 |
| Epochs | 70 |
| **Training** | |
| Permute Colours | Yes |
| Self-Play Eval Games | 5000 |
| **IPPO** | |
| **Optimization** | |
| Learning Rate | 0.0005 |
| Gamma Discount | 0.99 |
| GAE Lambda | 0.95 |
| Clip Epsilon | 0.2 |
| Entropy Coefficient | 0.01 |
| Value Function Coeff | 0.5 |
| Max Gradient Norm | 0.5 |
| Update Epochs | 4 |
| Num Minibatches | 4 |
| **Critic Network** | |
| Critic MLP | 512 |
| **Training** | |
| Total Timesteps | 2e10 |
| **Environments** | |
| Num Env Steps | 128 |
| Num Train Envs | 1024 |
| Num Eval Envs | 512 |
| Num Test Actors | 1024 |
| Num Test Envs | 512 |

In addition to the listed parameters, we consider setups with the following human data KL regularisation weights: 0.00, 0.01, 0.08, 0.13, 0.20, 0.30, 0.50, and 0.70. This range allows us to explore the effects of regularisation from a complete absence to a very strong influence on the learning process.

RESULTS AND DISCUSSION

We start by presenting the cross-play matrix in Figure 6.

From the cross-play matrix, we observe consistently high SP scores across all agents, indicating that all variations of HDR-IPPO lead to significant improvements over the baseline BC policy's mean SP score of 19.53. The HDR-IPPO agents with the lowest SP scores are those at the extremes of the regularisation spectrum: the non-regularised policy ($\lambda = 0.00$) and the one with the highest regularisation ($\lambda = 0.70$). The agent showing the best SP performance, with a mean score of 23.42, is the one trained with $\lambda = 0.13$.

This result provides initial empirical evidence supporting the effectiveness of the regularisation employed in HDR-IPPO. We hypothesise that excessively high regularisation weights might prevent

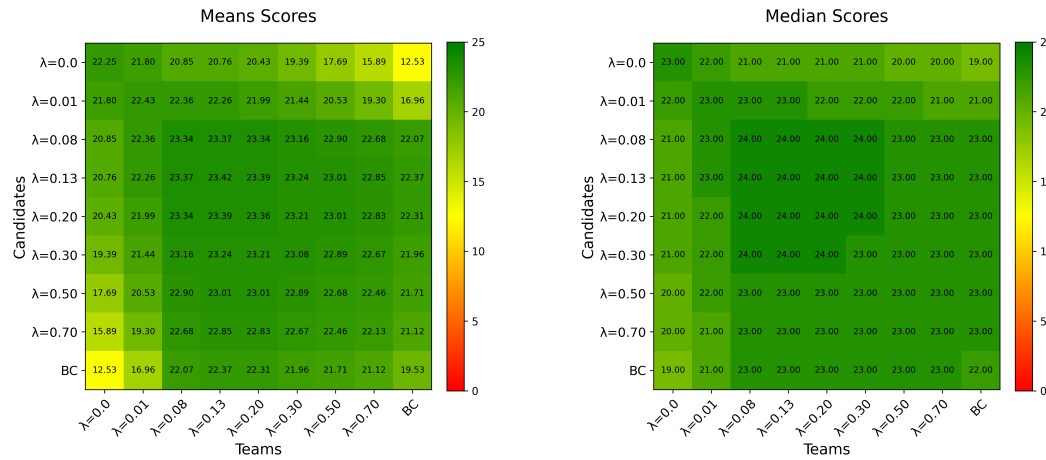

Figure 6: Cross-play performance matrix. Each element $(i, j)$ represents the average score achieved by a team comprising of Agent $j$ and Agent $i$, averaged over possible permutations of player positions. The average score for each team is calculated based on 25,000 games per permutation.

the policy to generalise, while the complete absence of regularisation could lead to the adoption of strategies that deviate significantly from human-like conventions.

Next, we examine the distribution of perfect and zero-score games. As shown in the Table 12 below, with an appropriate level of regularisation, we acquire a policy that achieves a high number of perfect games while completely eliminating games with a score of zero. In contrast, the non-regularised policy ($\lambda = 0.00$) yields only 143 perfect games, and its minimum score is comparable to that of the policy with $\lambda = 0.13$, which boasts 1599 perfect games. This discrepancy further supports our hypothesis that these agents employ fundamentally different strategies.

Table 12: Perfect and zero-score games count out of 5,000 SP games. We also show the minimum score in the set of 5,000 games.

|  | Perfect | Zero | Min |
|---|---|---|---|
| $\lambda = 0.00$ | 143 | 0 | 13 |
| $\lambda = 0.01$ | 247 | 0 | 14 |
| $\lambda = 0.08$ | 1351 | 0 | 14 |
| $\lambda = 0.13$ | 1599 | 0 | 13 |
| $\lambda = 0.20$ | 1821 | 0 | 15 |
| $\lambda = 0.30$ | 1564 | 3 | 0 |
| $\lambda = 0.50$ | 1280 | 21 | 0 |
| $\lambda = 0.70$ | 1166 | 113 | 0 |

Let us now analyse the cross-play results from the perspective of the BC policy. Examining the BC policy column in Figure 6, we observe that cross-play scores generally increase compared to the initial BC SP score when paired with most HDR-IPPO agents. However, a notable exception occurs for pairings with agents trained using $\lambda = 0.00$ and $\lambda = 0.01$. Despite these two policies having significantly higher SP scores than the BC policy, their coordination with the BC agent is poor, resulting in a substantial drop in cross-play performance. This widening gap between SP and cross-play scores is a recognised indicator of poor coordination. Additionally, the policy with the highest regularisation weight ($\lambda = 0.70$) does not exhibit this coordination breakdown, even though it performs worse in SP compared to the non-regularised agent ($\lambda = 0.00$). This observation provides empirical evidence that training with insufficient regularisation can lead to a divergence from the strategies encountered in the dataset used for BC, even when the final policy develops strong strategies and performs well in isolation.

Furthermore, let's examine the cross-play results from the perspective of the non-regularised policy ($\lambda = 0.00$). A clear trend emerges where cross-play performance deteriorates as the regularisation weight of the partner policy increases. This suggests that the non-regularised policy, having diverged from human-like conventions, struggles to coordinate effectively with agents that adhere more closely to those conventions. In contrast, for policies trained with higher regularisation weights, we observe the opposite trend. The gap between SP and cross-play scores diminishes, and in some cases, cross-play even yields higher scores than the weaker policy's SP performance. This indicates that these policies, guided by the KL regularisation term, converge towards similar strategies, enabling them to coordinate exceptionally well, even surpassing individual SP scores in certain instances. The same holds true when pairing these agents with the baseline BC agent, further underscoring their compatibility with human-like conventions. This analysis provides strong empirical evidence that policies trained with higher regularisation weights tend to converge to a shared set of strategies, and that those strategies align closely with ones learned during the BC procedure.

We now shift our focus to evaluating the agents on the held-out validation and test sets, as presented in Table 13. Increasing the regularisation weight generally leads to improved performance on the held-out data, suggesting better adherence to the conventions present in the training set. Notably, the increase from $\lambda = 0.00$ to $\lambda = 0.08$ results in a substantial improvement of over 20% accuracy on both the validation and test sets. Importantly, subsequent increases yield only marginal gains.

Importantly, agents that perform well in cross-play tend to have a lot stronger performance on the held-out datasets. Hence, we again show that they converge to similar conventions, but we also show that these conventions are very close to ones learned during training.

Table 13: Performance on hold out sets for all agents. We show the best result in **bold**.

|  | Val Loss | Val Acc | Test Loss | Test Acc |
|---|---|---|---|---|
| BC | 0.466 | **0.676** | **0.468** | **0.674** |
| $\lambda = 0.00$ | 7.368 | 0.330 | 7.385 | 0.327 |
| $\lambda = 0.01$ | 1.354 | 0.435 | 1.369 | 0.433 |
| $\lambda = 0.08$ | 0.716 | 0.574 | 0.725 | 0.571 |
| $\lambda = 0.13$ | 0.618 | 0.607 | 0.628 | 0.605 |
| $\lambda = 0.20$ | 0.540 | 0.636 | 0.548 | 0.634 |
| $\lambda = 0.30$ | 0.488 | 0.659 | 0.495 | 0.656 |
| $\lambda = 0.50$ | 0.475 | 0.665 | 0.481 | 0.663 |
| $\lambda = 0.70$ | **0.464** | 0.671 | 0.469 | 0.668 |

Finally, we turn our attention to the evolution of the KL divergence term throughout the training process. Figure 7 illustrates the KL divergence for all trained HDR-IPPO policies. It is immediately apparent that the KL terms for $\lambda = 0.00$ and $\lambda = 0.01$ are significantly higher than the others, rendering the remaining curves barely visible in the plot. This observation aligns with our previous findings, further reinforcing the notion that insufficient regularisation can lead to substantial divergence from the human-like strategies learned through BC.

To gain a clearer understanding of the KL divergence dynamics for policies with higher regularisation weights, we present Figure 8, which excludes the policies with $\lambda = 0.00$ and $\lambda = 0.01$. While policies with lower regularisation weights show increasing KL divergence during training, those with higher weights demonstrate a decreasing trend. This suggests that stronger regularisation effectively prevents the policy from deviating too far from the human-like strategies captured by the BC policy.

We do observe an initial increase in KL divergence for all policies during the first few update steps. This is likely attributable to the dominance of the IPPO loss over the KL divergence term in the early stages of training, particularly as the value function is being learned from scratch.

In conclusion, our ablation study provides compelling evidence that the KL regularisation term in HDR-IPPO effectively prevents divergence from the human-like strategies learned through BC. However, it is crucial to set the regularisation weight, $\lambda$, to a sufficiently large value to ensure sustained adherence to these conventions throughout the training process, particularly in the context of extended training duration.

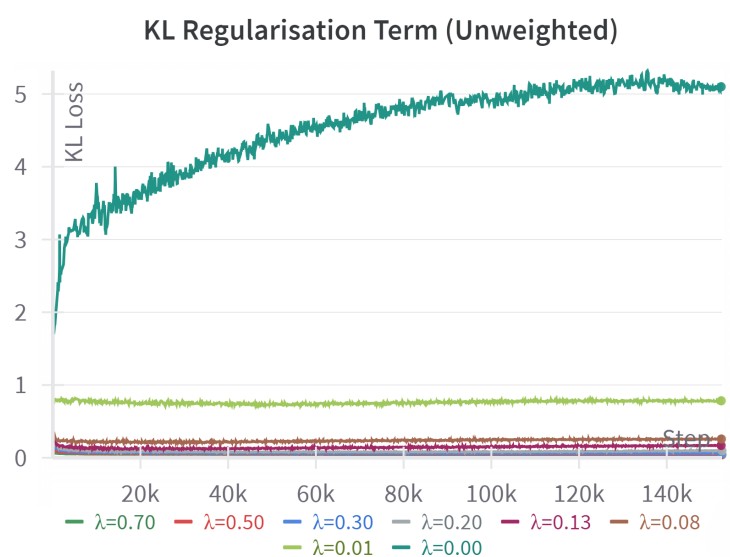

Figure 7: KL divergence throughout training for all trained HDR-IPPO policies.

## KL Regularisation Term (Unweighted)

Figure 8: KL divergence throughout training where $\lambda \geq 0.08$.

### A.8 BASELINES: HYPERPARAMETERS

For OBL, we use open-sourced weights and hyperparameters (Hu et al., 2021).

Table 14: Hyperparameters used for training BC, HDR-IPPO baselines on a 1,000-game data limit challenge. BC policies trained as baselines are used for starting points in HDR-IPPO and later for BR-BC.

| Hyperparameter | Two-Player Setting | Three-Player Setting |
|---|---|---|
| **Network Architecture** | | |
| Num Players | 2 | 3 |
| Activation | GELU | GELU |
| LSTM Layers | 512 | 512 |
| Input Embedding | 1024 | 512 |
| Decoder MLP | 256 | 256 |
| **BC** | | |
| **Optimisation** | | |
| Batch Size | 32 | 128 |
| Dropout | 0.5 | 0.5 |
| LR Schedule | Linear | Linear |
| Initial LR | 0.005 | 0.005 |
| Final LR | 0.0001 | 0.0001 |
| Epochs | 70 | 50 |
| **Training** | | |
| Permute Colours | Yes | Yes |
| Self-Play Eval Games | 5000 | 5000 |
| **IPPO during HDR-IPPO** | | |
| **Optimisation** | | |
| Learning Rate | 0.0005 | 0.0005 |
| Linear Schedule | True | True |
| Gamma Discount | 0.99 | 0.99 |
| GAE Lambda | 0.95 | 0.95 |
| Clip Epsilon | 0.2 | 0.2 |
| Entropy Coefficient | 0.001 | 0.001 |
| Value Function Coeff | 0.5 | 0.5 |
| Max Gradient Norm | 0.5 | 0.5 |
| Update Epochs | 4 | 4 |
| Num Minibatches | 4 | 4 |
| **Critic Network** | | |
| Critic MLP | 512 | 512 |
| **Training** | | |
| BC Policy KL Weight | 0.25 | 0.25 |
| Total Timesteps | 1e10 | 1e10 |
| **Environments** | | |
| Num Env Steps | 128 | 128 |
| Num Train Envs | 1024 | 1024 |
| Num Eval Envs | 512 | 512 |

Table 15: Hyperparameters used for training BC, HDR-IPPO baselines on a 5,000-game data limit challenge. BC policies trained as baselines are used for starting points in HDR-IPPO and later for BR-BC.

| Hyperparameter | Two-Player Setting | Three-Player Setting |
|---|---|---|
| **Network Architecture** | | |
|     Num Players | 2 | 3 |
|     Activation | GELU | GELU |
|     LSTM Layers | 512 | 512 |
|     Input Embedding | 1024 | 512 |
|     Decoder MLP | 256 | 256 |
| **BC** | | |
|   **Optimisation** | | |
|     Batch Size | 128 | 256 |
|     Dropout | 0.5 | 0.5 |
|     LR Schedule | Linear | Linear |
|     Initial LR | 0.005 | 0.005 |
|     Final LR | 0.0001 | 0.0001 |
|     Epochs | 50 | 50 |
|   **Training** | | |
|     Permute Colours | Yes | Yes |
|     Self-Play Eval Games | 5000 | 5000 |
| **IPPO during HDR-IPPO** | | |
|   **Optimisation** | | |
|     Learning Rate | 0.0005 | 0.0005 |
|     Linear Schedule | True | True |
|     Gamma Discount | 0.99 | 0.99 |
|     GAE Lambda | 0.95 | 0.95 |
|     Clip Epsilon | 0.2 | 0.2 |
|     Entropy Coefficient | 0.001 | 0.001 |
|     Value Function Coeff | 0.5 | 0.5 |
|     Max Gradient Norm | 0.5 | 0.5 |
|     Update Epochs | 4 | 4 |
|     Num Minibatches | 4 | 4 |
|   **Critic Network** | | |
|     Critic MLP | 512 | 512 |
|   **Training** | | |
|     BC Policy KL Weight | 0.25 | 0.25 |
|     Total Timesteps | 1e10 | 1e10 |
|   **Environments** | | |
|     Num Env Steps | 128 | 128 |
|     Num Train Envs | 1024 | 1024 |
|     Num Eval Envs | 512 | 512 |

Table 16: Hyperparameters used for training all IPPO and BR-BC baseline agents. Here, we use feed-forward architecture. BC agents used for BR-BC are the same as shown in Table 14 and 15, depending on the challenge variety.

| Hyperparameter | Two-Player Setting | Three-Player Setting |
|---|---|---|
| **Network Architecture** | | |
| Num Players | 2 | 3 |
| Activation | RELU | RELU |
| MLP | (512, 512) | (512, 512) |
| **IPPO** | | |
| **Optimisation** | | |
| Learning Rate | 0.0005 | 0.0005 |
| Gamma Discount | 0.99 | 0.99 |
| GAE Lambda | 0.95 | 0.95 |
| Clip Epsilon | 0.2 | 0.2 |
| Entropy Coefficient | 0.01 | 0.01 |
| Value Function Coeff | 0.5 | 0.5 |
| Max Gradient Norm | 0.5 | 0.5 |
| Update Epochs | 4 | 4 |
| Num Minibatches | 4 | 4 |
| **Critic Network** | | |
| Critic MLP | 512 | 512 |
| **Training** | | |
| Total Timesteps | 1e10 | 1e10 |
| **Environments** | | |
| Num Env Steps | 128 | 128 |
| Num Train Envs | 1024 | 1024 |
| **BR-BC** | | |
| BC Anneal Start | 1e9 | 1e9 |
| BC Anneal End | 6e9 | 6e9 |

