# OpenReview forum: "Ad-Hoc Human-AI Coordination Challenge"
_ICLR.cc/2025/Conference — Submitted to ICLR 2025_

### Official Review · Reviewer_4ENk · 2024-10-28

**Soundness:** 2
**Presentation:** 2
**Contribution:** 2
**Rating:** 3
**Confidence:** 4

**Summary:**

The paper is based on the Hanabi challenge, using human data to build evaluation partners and providing a benchmark along with a hosted evaluation platform to enable fair assessments.

**Strengths:**

The problem of ad hoc teamwork, which is closely related to human cooperation, is indeed worth attention, and Hanabi serves as an effective evaluation environment. I highly commend the authors for maintaining fairness by ensuring that evaluation partners are not public and are not solely determined by leaderboard rankings.

**Weaknesses:**

1. Low Effectiveness of Human Proxy

Ad hoc teamwork involves interacting with humans that the agents have never encountered before, posing a distributional shift generalization challenge, i.e, it is an special out-of-distribution challenge. Beyond the cost of collecting human data, the critical issue is whether the evaluation can reveal the algorithm’s ad hoc performance, i.e., whether the algorithm can maintain robust cooperation with a wide range deployment partners' policies.

Given that the human proxy is built through Behavior Cloning (BC) using limited human data, how can we ensure that these proxies introduce sufficiently diverse challenges? In other words, while the authors can guarantee that the human proxies are new partners (unseen during training), they cannot ensure diversity among these partners nor cover the full behavioral distribution of potential real-world human behaviors. This limitation weakens the validity of the evaluation.
I recommend referring to the following papers that discuss related issues:

Wang, Xihuai, Shao Zhang, Wenhao Zhang, Wentao Dong, Jingxiao Chen, Ying Wen, and Weinan Zhang. "ZSC-Eval: An Evaluation Toolkit and Benchmark for Multi-agent Zero-shot Coordination." arXiv preprint arXiv:2310.05208 (2023).

2. Insufficient Baselines – Missing Comparison with State-of-the-Art Methods

The currently provided baselines are mostly simple extensions of self-play, such as IPPO, BC, HDR-IPPO, and BR-BC. These baselines are clearly insufficient. In the field of ad hoc teamwork, Zero-Shot Coordination (ZSC) is often regarded as a special case of the ad hoc teamwork problem. The authors should consider incorporating ZSC algorithms into the benchmark.

Li, Yang, Shao Zhang, Jichen Sun, Yali Du, Ying Wen, Xinbing Wang, and Wei Pan. "Cooperative open-ended learning framework for zero-shot coordination." In International Conference on Machine Learning, pp. 20470-20484. PMLR, 2023.

Zhao, Rui, Jinming Song, Yufeng Yuan, Haifeng Hu, Yang Gao, Yi Wu, Zhongqian Sun, and Wei Yang. "Maximum entropy population-based training for zero-shot human-ai coordination." In Proceedings of the AAAI Conference on Artificial Intelligence, vol. 37, no. 5, pp. 6145-6153. 2023.

Yu, Chao, Jiaxuan Gao, Weilin Liu, Botian Xu, Hao Tang, Jiaqi Yang, Yu Wang, and Yi Wu. "Learning zero-shot cooperation with humans, assuming humans are biased." arXiv preprint arXiv:2302.01605 (2023).

Yan, Xue, Jiaxian Guo, Xingzhou Lou, Jun Wang, Haifeng Zhang, and Yali Du. "An efficient end-to-end training approach for zero-shot human-AI coordination." Advances in Neural Information Processing Systems 36 (2024).

Strouse, D. J., Kevin McKee, Matt Botvinick, Edward Hughes, and Richard Everett. "Collaborating with humans without human data." Advances in Neural Information Processing Systems 34 (2021): 14502-14515.

3. Hanabi is a Great Environment but Not Sufficient

While Hanabi is an excellent environment, it alone is insufficient to represent the diversity of scenarios involved in ad hoc teamwork. Ad hoc teamwork spans multiple scenarios, and using only Hanabi as the environment is not enough. The authors should consider integrating additional cooperative environments, such as Overcooked or Google Research Football, to enhance the benchmark.

4. Discussion of Benchmark Results

The authors only demonstrate through experiments that the constructed human proxy agents can play Hanabi with different partner agents.
As a benchmark paper, the authors should further analyze the results to help the community understand what new insights this Hanabi-based benchmark brings to the problem of ad hoc teamwork. Currently, the discussion only demonstrates that the proposed approach can compute benchmark results. I do not see further insights or novel contributions from the authors regarding this issue.

**Questions:**

1. How can the diversity of the generated human proxy agents be ensured?
2. I don’t fully understand how cross-play can verify that the human proxy behaves similarly to humans. A score alone cannot adequately demonstrate behavioral similarity. I hope the authors can provide further explanation.

---

> ### Author Response · Authors · 2024-11-19
> **Rebuttal**
>
> ## Rebuttal
>
> Thank you for your thoughtful review. We appreciate your commendation of our efforts to maintain fairness by ensuring evaluation partners are not public or solely determined by leaderboard rankings. Please find our response below:
>
> > _Weaknesses 1. and 2._
>
> Please see the general response.
>
> > _Weakness 3: Hanabi is a Great Environment but Not Sufficient:
> While Hanabi is an excellent environment, it alone is insufficient to represent the diversity of scenarios involved in ad hoc teamwork. Ad hoc teamwork spans multiple scenarios, and using only Hanabi as the environment is not enough. The authors should consider integrating additional cooperative environments, such as Overcooked or Google Research Football, to enhance the benchmark._
>
> While we acknowledge that ad-hoc teamwork spans diverse scenarios, many impactful studies focus on a single benchmark to provide insights. We chose Hanabi because it offers a complex, partially observable cooperative setting that's well-suited for studying ad-hoc teamwork.
>
> Regarding Overcooked, it's a fully observable environment, which doesn't align with our focus on partial observability challenges. As for Google Research Football, we currently lack the necessary human data to train human proxy agents, making it impractical for our benchmark at this time.
>
> > _Weakness 4: Discussion of Benchmark Results:
> The authors only demonstrate through experiments that the constructed human proxy agents can play Hanabi with different partner agents. As a benchmark paper, the authors should further analyze the results to help the community understand what new insights this Hanabi-based benchmark brings to the problem of ad hoc teamwork. Currently, the discussion only demonstrates that the proposed approach can compute benchmark results. I do not see further insights or novel contributions from the authors regarding this issue._
>
> Our main contribution is to make human-AI coordination evaluation more accessible by approximating evaluations with humans at a lower cost using human proxy agents. This approach brings human-AI coordination studies closer to real-world scenarios without the logistical challenges of involving human participants in every evaluation.
>
> If there are specific analyses or further insights you believe would provide additional value, please let us know.
>
> > _Question 1: How can the diversity of the generated human proxy agents be ensured?_
>
> See the general response.
>
> > _Question 2: I don’t fully understand how cross-play can verify that the human proxy behaves similarly to humans. A score alone cannot adequately demonstrate behavioral similarity. I hope the authors can provide further explanation._
>
> We agree that achieving similar scores in cross-play is not sufficient **on its own** to verify that the human proxy behaves humans. To provide a more comprehensive demonstration of behavioural similarity, we have conducted additional analyses beyond score comparisons. We have included the cross-entropy loss for action prediction which measures how well the human proxy can predict the actions of real human players based on game states. We show that human proxies have cross-entropy scores close to BC policies that we know follow human-like strategies, given they are trained on human data only.
>
> Additionally, we have analysed behavioural metrics in the updated version of the paper, following the approach proposed by Canaan et al. (2020). Our results show that these metrics are nearly identical between the human proxy agents and the human gameplay data.
>
> By combining cross-play evaluations with these analyses, we provide evidence that our human proxies not only achieve have strong performance but also exhibit human-like behaviour.
>
> Finally, we would like to clarify that our methodology for creating human proxy agents is not novel but builds upon established techniques from previous research. Specifically, we utilise BC combined with regularised RL, methodologies that have been extensively explored in prior works such as Cornelisse et al. (2024), Jacob et al. (2022), and Bakhtin et al. (2021).
>
> ## Conclusion
>
> Thank you again for your thorough review. We hope that the reviewer feels we have addressed their questions and welcome any further discussion. We also ask that, if all their concerns are met, the reviewer consider increasing their support for our paper.
>
>
> ---
> Canaan et al. (2020). Generating and Adapting to Diverse Ad-Hoc Cooperation Agents in Hanabi. ArXiv. https://doi.org/10.1109/TG.2022.3169168
>
> Cornelisse et al. (2024). Human-compatible driving partners through data-regularized self-play reinforcement learning. ArXiv. https://arxiv.org/abs/2403.19648
>
> Jacob et al. (2021). Modeling Strong and Human-Like Gameplay with KL-Regularized Search. ArXiv. https://arxiv.org/abs/2112.07544
>
> Bakhtin, et al. (2022). Mastering the Game of No-Press Diplomacy via Human-Regularized Reinforcement Learning and Planning. ArXiv. https://arxiv.org/abs/2210.05492

---

> > ### Comment · Reviewer_4ENk · 2024-11-20
> > **Discussion of Benchmark Results**
> >
> > Regarding the discussion on the benchmark results, I would like to add that a meaningful benchmark paper should demonstrate the necessity of the evaluation and discuss the implications of the evaluation results, such as the insights they bring to the field as a whole. At present, the authors merely present the evaluation results, and I fail to see how this benchmark addresses the shortcomings of current ad hoc problem evaluations. Employing a better method for generating human proxy is not sufficient to justify a benchmark paper as offering new insights to the field, particularly when it fails to raise any issues worthy of discussion or further exploration. The authors should reflect further on the implications of the current benchmark results for future ad hoc problems. This is crucial for establishing why future ad hoc algorithms would need to rely on this benchmark.

---

> > > ### Author Response · Authors · 2024-11-23
> > > **Discussion Section**
> > >
> > > Firstly, we thank you for your valuable feedback and for taking the time to review our work. We have now added a discussion paragraph analysing the benchmark results. Our work addresses a critical gap in the field: while many papers include human-AI evaluation, these evaluations typically rely on proprietary datasets and closed-source human proxies. This makes reproducing results and comparing methods impossible. Moreover, we specifically designed the limited data challenge to mirror a crucial real-world scenario, given that we provide the first open-source Hanabi human play dataset: algorithms must learn to cooperate with a specific target population while having access to only a small training dataset (1000 games) from that population. This reflects common practical constraints where collecting large-scale human data is infeasible, making the benchmark particularly relevant for developing robust human-AI coordination methods.
> > >
> > > Additionally, please see the general response where we addressed diversity issues and continued the discussion.

---

### Official Review · Reviewer_phEU · 2024-11-03

**Soundness:** 3
**Presentation:** 3
**Contribution:** 3
**Rating:** 6
**Confidence:** 5

**Summary:**

This paper proposes to use Hanabi as an ad-hoc human-ai collaboration and sets up an evaluation methodology. The authors train behavior cloning (BC) / BC regularized agents of humans as a low-cost replacement for human players. The authors also promise a release of small subsets of human play data and host an online benchmark.

**Strengths:**

In general, I think the paper is well-written. Although using hanabi as a human-ai collaboration benchmark has been explored by several groups, I believe the effort promised in this paper will significantly contribute to the multi-agent ad hoc human-ai collaboration research.

**Weaknesses:**

I believe this paper lacks a justification/study on why the trained human proxy is a fair judge for this challenge.
(1) The bc policy, if not treated well in latent representation, eventually converges to a single average human policy. So BC+RL hypothetically stays around this average policy. One important aspect of collaborating humans in that human may demonstrate diverse collaboration conventions. Looking at Table 6, I wonder if with specific network mechanism to capture human behavior diversity will help the accuracy.

(2) As a human-ai benchmark, shouldn’t the qualitative/quantitative evaluation against real humans still be necessary for the public leaderboard? Then, you can compare the correlation between play with humans and play with human proxy. Otherwise, I believe the authors should spend more time justifying why human-proxy=human.

**Questions:**

See the question in weakness.

(+1) How does the performance of BC/BC+RL trained on a small released subset compare to that of BC/BC+RL trained on the entire dataset, and what causes this difference? Is it due to the training not converging, or does it suggest a divergence in convention distributions between the two?

---

> ### Author Response · Authors · 2024-11-19
> **Rebuttal**
>
> ## Rebuttal
>
> Thank you for your thorough and insightful review. We are especially grateful for your positive feedback, noting that our paper is well-written and has the potential to make a significant contribution to the multi-agent ad hoc human-AI collaboration research. Your comments are greatly appreciated and motivate us to ensure clarity in addressing your concerns.
>
> > _(1)_
>
> Please see the general response.
>
> > _(2): As a human-ai benchmark, shouldn’t the qualitative/quantitative evaluation against real humans still be necessary for the public leaderboard? Then, you can compare the correlation between play with humans and play with human proxy. Otherwise, I believe the authors should spend more time justifying why human-proxy=human._
>
> Thank you for raising this important point. While we discuss this issue in our general response, we'd like to provide additional thoughts here. We agree that evaluating agents against real humans would be ideal for a human-AI benchmark. However, conducting large-scale evaluations with human participants is often impractical due to logistical challenges and resource constraints.
>
> To address this, we built human proxies using a methodology that has been extensively studied in prior research. It has been shown that regularised RL leads to policies that are compatible with the social conventions of the human reference group. Additionally, we have also conducted studies to validate that our human proxies exhibit human-like qualities found in the dataset.
>
> Thank you again for your insightful feedback. We hope this explanation clarifies our approach and reassures you of the robustness of our human proxies in the benchmark.
>
> > _(+1): How does the performance of BC/BC+RL trained on a small released subset compare to that of BC/BC+RL trained on the entire dataset, and what causes this difference? Is it due to the training not converging, or does it suggest a divergence in convention distributions between the two?_
>
> Thank you for a great question. We include the performance of BC (Behavioral Cloning) and BC+RL (Behavioral Cloning with Reinforcement Learning) when trained on a small released subset versus the entire dataset in our paper, specifically in Table 2 and Table 4.
>
> In the two-player setting with a 1,000-game limit, BC achieves a mean self-play score of only 2.12. This low score suggests that BC struggles to learn how to navigate diverse game states from such a small dataset since the limited data doesn't cover a wide range of game states.
>
> However, similar is observed even when BC is trained on a large-scale dataset, where the mean score is relatively high. BC agents seem to either achieve strong scores or "bomb-out" by losing all lives, a phenomenon we discuss in Section 5.1. In contrast, BC+RL agents show improved performance even when trained on the smaller subset. They don't exhibit the same tendency to lose all lives, suggesting that the RL component helps them adapt better to new situations. However, we observed that during the RL training loop, the score plateaus. We hypothesise that this plateau occurs because the KL-divergence regularisation term limits significant policy updates, preventing the agent from deviating too far from the initial BC policy.
>
> In summary, BC agents trained with the full dataset generalise better and achieve higher scores. The small subset doesn't provide this richness, so agents trained solely on it can't match the performance of those trained on the entire dataset. We hope this clarifies the differences in performance and their causes. If you have further questions or need additional details, please let us know.
>
> ## Conclusion
>
> Thank you again for your thorough review. We hope that the reviewer feels we have addressed their questions and welcome any further discussion. We also ask that, if all their concerns are met, the reviewer consider increasing their support for our paper.

---

> > ### Comment · Reviewer_phEU · 2024-12-03
> >
> > I read all the reviews and comments, and I believe the authors have presented a better qualitative analysis regarding human proxies, which was a common concern during the rebuttal.
> >
> > After reflecting on what the benchmark could bring and checking the results at ah2ac2.com, I found that the result was a bit unexpected because OBL (which is an algorithm that encourages policies to converge to the same coordination across training) ranked first.
> >
> > I might also need a little bit of clarification on whether and to what extent interaction history from previous episodes with the human proxy is available to participant algorithms. This information is critical, as it underpins the fundamental assumptions of many ad hoc teamwork algorithms, different from ZSC algorithms.
> >
> > Lastly, (not affecting my ratings because it is not a claimed contribution), if the benchmark could enable human participation—for instance, allowing random online players to challenge AI systems—its value could be significantly enhanced. Incorporating an Elo rating system to estimate coordination ability might provide a meaningful metric for such interactions.

---

> > > ### Author Response · Authors · 2024-12-03
> > >
> > > We thank reviewer phEU for engaging in the discussion and appreciate the acknowledgement of our efforts to address the reviewers' concerns.
> > >
> > > > After reflecting on what the benchmark could bring and checking the results at ah2ac2.com, I found that the result was a bit unexpected because OBL (which is an algorithm that encourages policies to converge to the same coordination across training) ranked first.
> > >
> > > We were not surprised by OBL's top ranking. OBL is specifically designed to coordinate effectively with human-like partners, aligning closely with models of human communication behaviour. The authors of OBL highlight several points that explain its strong performance:
> > >
> > > - _"Unlike SP where conventions are formed implicitly as each agent adapts to the random biases of its partner, OBL exhibits a consistent path of evolution that can be understood by reasoning through the behaviors of the lower level OBL policy. The final level of OBL is the most similar to Clone Bot in how it decides to play cards."_
> > >
> > > - _"OBL (level 1) already cooperates with human-like policies in Hanabi (proxied by CloneBot) better than any prior method."_
> > >
> > > -  _"We train three additional levels and the evaluation results are shown in Table 1. OBL level 2 infers beliefs over unseen
> > >  cards assuming other agents are more likely to take good actions and to share factual information about useful cards the way OBL level 1 would, and so on for higher levels. Since other agents, even if they have unknown conventions, will also often act this way, these beliefs are sufficiently robust that OBL level 2 achieves even better scores than OBLlevel 1 with Clone Bot and with all other tested agents."_
> > >
> > > - _"OBL is conceptually related to the Rational Speech Acts (RSA) framework, which has been able to model and explain a large amount of human communication behavior (Frank & Goodman, 2012). RSA assumes a speaker-listener setting with grounded hints and starts out with a literal listener (LL) that only considers the grounded information provided. Like CH, RSA then introduces a hierarchy of speakers and listeners, each level defined via Bayesian reasoning (i.e., a best response) given the level below. OBL allows us to train an analogous hierarchy and thus introduce pragmatic reasoning in a controlled fashion."_
> > >
> > > - _"OBL’s performance gains under ZSC directly translate to state-of-the-art ad-hoc teamplay and human-AI coordination results, validating the “ZSC hypothesis”_
> > >
> > > These insights demonstrate that OBL's design inherently promotes effective coordination with human-like policies. Its inclusion in our benchmark was based on this expectation, and to the best of our knowledge, it (and its variants) remains SOTA in this setting.
> > >
> > > > I might also need a little bit of clarification on whether and to what extent interaction history from previous episodes with the human proxy is available to participant algorithms. This information is critical, as it underpins the fundamental assumptions of many ad hoc teamwork algorithms, different from ZSC algorithms.
> > >
> > > Participants have access to the full interaction history available during evaluation and to past trajectories. Specifically, they can observe the entire game from the perspective of the agent(s) they control, including the actions of the human proxy. While we do not provide the observation used by our proxies to make decisions, participants can reconstruct the game state and observations based on the available information. It would indeed be interesting to see participants leverage state reconstruction in their algorithms. Given that we focus on partially observable settings and provide a small sample of games that can be used during training, we believe this is the fair way to conduct evaluation.
> > >
> > > > Lastly, (not affecting my ratings because it is not a claimed contribution), if the benchmark could enable human participation—for instance, allowing random online players to challenge AI systems—its value could be significantly enhanced. Incorporating an Elo rating system to estimate coordination ability might provide a meaningful metric for such interactions.
> > >
> > > After the challenge concludes—to keep everything fair—we plan to provide full access to our human proxies, including an interface that allows human players to interact with them directly. We love the idea of incorporating an Elo rating system to estimate coordination ability, and we'll definitely explore this further.
> > >
> > > We once again thank the reviewer for engaging in the discussion! If we have addressed all the issues, we would kindly ask the reviewer to increase their support for our paper!

---

> ### Comment · Area_Chair_wSdm · 2024-11-25
> **Please read rebuttal**
>
> Dear Reviewer phEU, Could you please read the authors' rebuttal and give them feedback at your earliest convenience? Thanks. AC

---

### Official Review · Reviewer_c4nE · 2024-11-03

**Soundness:** 2
**Presentation:** 3
**Contribution:** 2
**Rating:** 5
**Confidence:** 5

**Summary:**

This paper proposes an Ad Hoc Human-AI Coordination Challenge (AH2AC2) to promote research on seamless coordination between AI agents and humans in the Hanabi benchmark environment. Recognizing the challenges of evaluation with human partners, the authors propose human agents using behavioral cloning based on a large human game dataset and reinforcement learning techniques. They introduce two data-limited challenge settings (1,000 and 5,000 games) to allow agents to exploit limited human data. To prevent overfitting, the authors adopt an evaluation protocol of hosted agents and maintain a public leaderboard for community tracking.

**Strengths:**

Originality: This paper addresses the important problem of human-AI coordination, which is a challenging and timely area for practical applications. Hanabi is a mature environment for exploring these dynamics, and AH2AC2 appears to bring structure to the field through a well-thought-out evaluation protocol.

Quality: The development of a human agent and the curation of a large-scale dataset of human Hanabi gameplay (over 10,000 games) demonstrate a significant amount of engineering effort and careful design for reproducibility. The provided public leaderboard adds value by creating a community-driven benchmark for progress.

Clarity: The paper is well-structured, with a clear description of each component, from the human agent to the baseline methods and evaluation protocols.

Significance: Human-AI coordination is a relevant problem, and the AH2AC2 challenge has the potential to be a useful testbed for researchers to evaluate methods in a reproducible manner without a human partner, thereby addressing a practical gap in the field.

**Weaknesses:**

Based on my understanding, this paper primarily serves as a benchmark paper, introducing a new benchmark based on a human dataset. However, several key weaknesses limit its impact:

	1.	Insufficient Analysis of Human Dataset and Proxy Agents: The paper lacks an in-depth analysis of the human dataset and human proxy agents. In Table 1, the authors provide only basic statistics (min, max, average, median, and standard deviation) for scores and game lengths. A more detailed exploration of the dataset, including aspects like trajectory, strategy diversity, and behavioral patterns, would greatly enhance the depth of the analysis and provide better insights into the proxies’ fidelity.

	2.	Incomplete Introduction to the Challenge: The challenge setup is not thoroughly introduced, leaving the structure and implementation of the leaderboard unclear. Without this information, it is difficult to assess the challenge’s usability and rigor. The lack of leaderboard details also raises concerns that the development of the challenge may be incomplete.

	3.	Figure 3 is confusing, especially in the two-player setting, where the performance of all compositions appears similar. This could be due to multiple reasons—such as a highly effective dataset that results in nearly perfect models even using BC. However, this raises questions about the benchmark’s ability to differentiate between varying model qualities. Furthermore, it is unclear what the three-player setting represents and how it differs from the two-player configuration.

	4.	Limited and Outdated Baselines: Given that this is a benchmark paper, the choice of baselines is crucial. The limited and outdated baselines evaluated here fail to showcase the challenge’s full potential. Including a broader range of up-to-date baselines would allow for a more comprehensive comparison and strengthen the benchmark’s value to the community.

**Questions:**

See weakness.

---

> ### Author Response · Authors · 2024-11-19
> **Rebuttal**
>
> ## Rebuttal
>
>
> Thank you for your thorough and constructive review. We deeply appreciate your recognition of our work's originality in addressing human-AI coordination, the quality of our dataset and engineering efforts, the clarity of our presentation, and the significance of AH2AC2 as a practical testbed for the field. Please find our responses below.
> > _1._
>
> Please see the general response. We clarify and provide additional metrics that are now available in the appendix.
>
> > _2. Incomplete Introduction to the Challenge: The challenge setup is not thoroughly introduced, leaving the structure and implementation of the leaderboard unclear. Without this information, it is difficult to assess the challenge's usability and rigour. The lack of leaderboard details also raises concerns that the development of the challenge may be incomplete._
>
> Thank you for highlighting the need for more clarity on the challenge setup. In response, we have added additional details about the challenge and its structure to the appendix. Our supplementary material includes the code used for evaluating the baselines, along with the open-sourced weights for these models.
>
> While our original plan was to make the leaderboard public upon the official release of the challenge, we understand your concerns about its availability. To address this, we have decided to release the leaderboard now, accessible at ah2ac2.com. We hope this helps reduce any uncertainties and brings us a step closer to a full public release and access to human proxies.
>
> > _3. Figure 3 is confusing, especially in the two-player setting, where the performance of all compositions appears similar. This could be due to multiple reasons—such as a highly effective dataset that results in nearly perfect models even using BC. However, this raises questions about the benchmark's ability to differentiate between varying model qualities. Furthermore, it is unclear what the three-player setting represents and how it differs from the two-player configuration._
>
> Thank you for your comments regarding Figure 3. We understand that the performance similarities in the two-player setting may appear confusing. In this setting, Behavioral Cloning (BC) agents achieve decent scores (18.95 and 19.66 in self-play evaluation). However, our human-proxy agents significantly improve these results to 22.56 and 23.02, which is a substantial enhancement. It's important to note that in Hanabi, moving from scores below 20 to above 22 is quite challenging; achieving decent scores is relatively straightforward, but attaining near-perfect scores requires sophisticated coordination strategies and very little mistakes.
>
> Additionally, as shown in Table 2, BC agents perform poorly out of distribution, often resulting in failed games. In contrast, human-proxy agents maintain robust performance. The difference between the human-proxy and BC agents is even more pronounced in the three-player setting, highlighting the advantages of our method.
>
> Regarding the three-player setting, it involves three players participating in the game, unlike the two-player configuration. We included both configurations in our evaluation to provide a comprehensive benchmark. The three-player setup is indeed more difficult and is often overlooked in previous works on Hanabi, whether in zero-shot coordination (ZSC) or ad-hoc settings. By addressing both, we aim to cover a broader spectrum of gameplay scenarios.
>
> We appreciate your attention to these details and your valuable feedback. If any part of our explanation remains unclear, please let us know, and we would be happy to provide further clarification.
>
> > _4._
>
> Please see the general response. We provide a discussion on the topic and add an additional baseline.
>
> ## Conclusion
> We sincerely thank you for your detailed review and the recognition of our paper's strengths.
> We hope our responses have adequately addressed your concerns.
> We believe that with these clarifications and improvements, our work aligns well with the positive aspects you've highlighted in terms of originality, quality, clarity, and significance.

---

> ### Comment · Area_Chair_wSdm · 2024-11-25
> **Please read rebuttal**
>
> Dear Reviewer c4nE, Could you please read the authors' rebuttal and give them feedback at your earliest convenience? Thanks. AC

---

> ### Comment · Reviewer_c4nE · 2024-11-26
>
> Thank you for your response. I will increase my score to 5 to acknowledge your effort. However, I still believe the paper is below the acceptance threshold (especially the novelty) and would benefit from further improvements in the future.

---

> > ### Author Response · Authors · 2024-11-27
> > **Clarification Request on Reviewer's Comment**
> >
> > Thank you for the updated review and for acknowledging the effort we have put into addressing the concerns raised during the rebuttal period and the increase in your score to 5. However, we are somewhat puzzled by the assertion that our paper is "below the acceptance threshold (especially the novelty)" given that the original review highlighted "originality" and "significance" as strengths.
> >
> > As a benchmark paper, our work is intended to provide a new evaluation framework, dataset, and challenge for the research community. Given that our paper introduces a first-ever structured evaluation protocol for human-AI coordination in the complex Hanabi environment, along with the first-ever released dataset of human gameplay and standardised human proxies, we believe that our work is indeed novel and contributes significantly to the field.
> >
> > We would appreciate it if you could clarify what specific aspects of novelty you believe are lacking in our submission. Is it the case that the reviewer expects additional methodological innovations beyond the establishment of the benchmark? If so, we would argue that this expectation may not be appropriate for a benchmark paper, which serves a different purpose in the research ecosystem.
> >
> > Also, given that we have addressed your initial concerns, what specific weaknesses prevent this paper from meeting the acceptance threshold? We are committed to advancing the field of human-AI coordination and your detailed feedback would help us understand any remaining reservations.
> >
> > If there are still concerns that prevent the paper from reaching the acceptance threshold, we kindly ask that the reviewer specify these concerns clearly and provide us with actionable feedback. If we addressed the weaknesses raised in the original review, we respectfully request that the reviewer reconsider their assessment.

---

### Official Review · Reviewer_JsHP · 2024-11-03

**Soundness:** 2
**Presentation:** 2
**Contribution:** 1
**Rating:** 5
**Confidence:** 3

**Summary:**

I acknowledge the rebuttal from the authors and and happy to rains my score.

 -------------------
This paper proposes a methodology for evaluating human-AI coordination in Hanabi using human-like proxy agents. The authors introduce the Ad-Hoc Human-AI Coordination Challenge (AH2AC2) with two sub-challenges that utilize limited amounts of human gameplay data (1,000 or 5,000 games). They contribute a new open-source Hanabi human gameplay dataset and baselines for both sub-challenges, including zero-shot coordination methods and methods leveraging the limited human data.

**Strengths:**

1.	The paper addresses the important issue of evaluating human-AI coordination in imperfect information settings.
2.	Conducting human proxy for evaluation is potentially valuable for ad-hoc teamwork.
3.	The release of a new Hanabi human gameplay dataset is a positive contribution.
4.	The challenge focusing on limited human data is relevant for real-world applications.

**Weaknesses:**

1. The core idea of employing behavioral cloning to create human-like agents in Hanabi builds upon previous work (Carroll et al., 2019), lacking significant innovation.
2.	The paper neglects to compare against more recent advancements in human-AI coordination for Hanabi and some interesting related works in overcooked.
3.	While the human dataset and proxy models are a valuable contribution, the paper lacks a comprehensive analysis of the dataset's diversity and the strategies exhibited by the proxy agents. It would be beneficial to delve deeper into the dataset and compare it with existing analyses of human behavior in Hanabi.
4.	A demo website or interactive platform for the challenge would enhance accessibility and engagement with the community. Is it possible to release all datasets? If not, why? Can you release the pipeline of the dataset preprocess?

**Questions:**

see weakness

---

> ### Author Response · Authors · 2024-11-19
> **Rebuttal**
>
> ## Rebuttal
>
> Thank you for your thorough and constructive review. We particularly appreciate your recognition of our work's value in addressing human-AI coordination evaluation, the potential of human proxies for ad-hoc teamwork, and the importance of our dataset contribution and limited-data challenge for real-world applications. Please find our response to your comments below:
>
> > _1. The core idea of employing behavioral cloning to create human-like agents in Hanabi builds upon previous work (Carroll et al., 2019), lacking significant innovation._
>
> There appears to be a misunderstanding about the methodology used for training human proxies. We do not use BC to create human proxies – instead, we employ regularised RL, an approach developed very recently for building human-like partners in partially observable environments (Cornelisse et al. 2024, Jacob et al. 2022, Bakhtin et al. 2021).
>
> Also, our paper's primary contribution/innovation isn't in developing new methods for building human-like agents, but rather in delivering the following contributions to the field:
> - We are the first to provide strong, validated and reproducible human proxies for standardised human-AI coordination evaluation.
>    - This addresses a significant gap in existing work, which relies either on closed-source BC agents trained on private datasets or a very small pool of human players.
> - We contribute the first-ever open-source dataset of human Hanabi gameplay.
>    - We also use this dataset to formalise the human-AI coordination challenge with limited data.
> - We establish the evaluation protocol for ad-hoc human-AI coordination in partially observable environments
>    - This protocol ensures fair and reproducible evaluation across different approaches.
>
> A summary of all key contributions can be found in the paper and given that these are acknowledged as strengths of our work in the review, we are surprised by the contribution score and would appreciate clarification.
>
> > _2. and 3._
>
> We address these issues in our global response.
>
> > _4. A demo website or interactive platform for the challenge would enhance accessibility and engagement with the community. Is it possible to release all datasets? If not, why? Can you release the pipeline of the dataset preprocess?_
>
> Thank you for these constructive suggestions. A demo website is indeed an excellent idea. Currently, all open-sourced human games can be visualised using the JaxMARL package, providing immediate accessibility to researchers and challenge participants. We will look into integrating our agents into the hanab.live platform if there is interest.
>
> Regarding demo access to human proxies and/or their training data, we've made careful design choices to maintain the integrity of AH2AC2. While we aim to release these resources in the future, providing open access now could compromise the fairness of the evaluation procedure. We have developed a server that provides controlled access to human proxies, ensuring standardised and reproducible evaluation for all participants.
>
> On the data preprocessing pipeline, we want to clarify that our repository already includes all code for data loading, augmentation, and transformations – the same pipeline used to train our baseline agents and human proxies. The dataset we release undergoes minimal preprocessing, where we only remove corrupted games (unfinished or abandoned) and games that use Hanabi variants. We don't perform any additional filtering.
>
>
> ## Conclusion
> Thank you again for your thorough review. We hope that the reviewer feels we have addressed their questions and clarified our work and welcome any further discussion. We also ask that, if all their concerns are met, the reviewer consider increasing their support for our paper.
>
> ---
> Canaan et al. (2020). Generating and Adapting to Diverse Ad-Hoc Cooperation Agents in Hanabi. ArXiv. https://doi.org/10.1109/TG.2022.3169168
>
> Cornelisse et al. (2024). Human-compatible driving partners through data-regularized self-play reinforcement learning. ArXiv. https://arxiv.org/abs/2403.19648
>
> Jacob et al. (2021). Modeling Strong and Human-Like Gameplay with KL-Regularized Search. ArXiv. https://arxiv.org/abs/2112.07544
>
> Bakhtin, et al. (2022). Mastering the Game of No-Press Diplomacy via Human-Regularized Reinforcement Learning and Planning. ArXiv. https://arxiv.org/abs/2210.05492

---

> ### Comment · Area_Chair_wSdm · 2024-11-25
> **Please read rebuttal**
>
> Dear Reviewer JsHP, Could you please read the authors' rebuttal and give them feedback at your earliest convenience? Thanks. AC

---

### Author Response · Authors · 2024-11-19
**Rebuttal**

We thank all reviewers for the time taken to review the submission and for the constructive feedback. We address the common points raised by multiple reviewers in this general response.

## Analysis of the Dataset and the Human Proxies
*Addressing concerns raised by reviewers JsHP and c4nE regarding dataset analysis and proxy agents.*

We reiterate that our dataset contains games using H-group convention strategies ([H-Group Conventions](https://hanabi.github.io/learning-path/)).

We have now added standard behavioural metrics proposed by Canaan et al., 2020 to the appendix. Specifically, we include Information Per Play (IPP) and Communicativeness. Our analysis shows that these metrics are nearly identical between human proxy rollouts and dataset trajectories.

We welcome specific suggestions and actionable feedback for additional analysis. We would appreciate it if reviewer JsHP could provide references to the _"existing analyses of human behaviour in Hanabi"_ mentioned in their review.

## Human Proxy Diversity & Fidelity
*Addressing concerns raised by reviewers phEU and 4ENk regarding human proxy evaluation and diversity.*

The evaluation of ad-hoc coordination fundamentally depends on choosing appropriate test-time policies. While some approaches, like Adversity proposed by Cui et al., 2023, focus on generating diverse policies with different conventions, our work takes a different approach. Instead of optimising for coordination with arbitrary diverse policies, we focus on coordination with a specific population of human(-like) players that are inherenlty meaningful.

Even if our policies converge toward an average human policy found in the dataset, this policy still encompasses multiple human-like strategies. For context, even the basic H-convention level requires mastery of multiple strategies. Additionally, H-convention strategies build upon each other, creating a rich space of independent strategies.

While we acknowledge that our proxies don't cover the full spectrum of potential human behaviours, *they represent the first standardised and cheap human proxies for evaluation*. This is a significant advance over existing approaches in human-AI coordination research:
- Adversity: Undocumented clone bot trained with supervised learning on hidden data.
- OBL: Undocumented clone bot trained with supervised learning on hidden data.
- K-level Hierarchies: Uundocumented clone bot trained with supervised learning on hidden data.
- OT-OBL: Undocumented clone bot trained with supervised learning on hidden data.
- Learning Intuitive Policies: Only 10 human players.
- Other-Play (OP): Only 20 human players with basic Hanabi familiarity.

Our approach for building human proxies is more sophisticated than any of the previous approaches used within Hanabi, combining BC and regularised RL (Cornelisse et al., 2024, Jacob et al., 2022, Bakhtin et al., 2021).

## Insufficient Baselines
*Addressing concerns raised by reviewers JsHP, c4nE, and 4ENk regarding baseline comparisons.*

We acknowledge the reviewers' suggestions for additional baseline comparisons and are taking action to address this feedback. While algorithms suggested by 4ENk are promising and interesting, some are not applicable in our setting - methods have only been validated/implemented/defined in *two-player* Overcooked – a *fully observable environment* that is substantially simpler than partially observable Hanabi.

To illustrate:
- COLE: designed for two player settings only.
- E3T: not applicable to partially observable setting - it employs a partner prediction module.
- HSP: requires domain knowledge to design a suitable set of events.
- All the methods are defined only in the two-player setting, algorithm for the n-player setting isn't clearly defined.

Nevertheless, we do appreciate meaningful feedback and agree with the reviewers that we should include additional baselines. We add OP since it is ZSC method similar in spirit to population-based methods - it constructs a symmetry-augmented population for agents to be robust with. Results are presented in the paper for two-player setting and incoming for the three-player setting. Also, we are currenlty running FCP and will update the paper as soon as possible.

---

Canaan et al. (2020). Generating and Adapting to Diverse Ad-Hoc Cooperation Agents in Hanabi. ArXiv. https://doi.org/10.1109/TG.2022.3169168

Cui et al. (2023). Adversarial Diversity in Hanabi. In The Eleventh International Conference on Learning Representations.

Cornelisse et al. (2024). Human-compatible driving partners through data-regularized self-play reinforcement learning. ArXiv. https://arxiv.org/abs/2403.19648

Jacob et al. (2021). Modeling Strong and Human-Like Gameplay with KL-Regularized Search. ArXiv. https://arxiv.org/abs/2112.07544

Bakhtin, et al. (2022). Mastering the Game of No-Press Diplomacy via Human-Regularized Reinforcement Learning and Planning. ArXiv. https://arxiv.org/abs/2210.05492

---

> ### Comment · Reviewer_4ENk · 2024-11-20
>
> 1.Diversity Problem
>
> I disagree with the authors’ statement regarding diversity. If the paper focuses solely on a limited and narrow range of human behaviors, it will severely restrict the evaluation’s effectiveness and applicability. How should evaluations be conducted if future ad hoc algorithms are not optimized for the range of human behaviors considered in this work? Would the authors’ benchmark be inapplicable in such cases? I believe that an effective ad hoc evaluation should prioritize diversity to assess which behaviors, across the widest possible distribution, an agent can adapt to and cooperate with, rather than merely providing a specific and narrow behavior distribution.
>
> 2. Human Data
>
> Additionally, please be cautious when using the phrase “the first standardized and cheap human proxies for evaluation.” To my knowledge, the use of human proxies in Overcooked has already become quite widespread. Reconstructing such proxies in Hanabi does not constitute the first instance of building human proxies in human-AI coordination. I am also puzzled by the authors’ claim that using human data to construct proxies is a “cheap” approach. Such a huge human dataset cannot be seen as a cheap solution. For example, the authors mention that the absence of human data in GRF prevents evaluation. If there were a method to construct evaluation partners without relying on real human trajectories, would that not fundamentally resolve this issue?

---

> > ### Author Response · Authors · 2024-11-20
> >
> > First off -- thank you for engage in the discussion. I believe there are a couple of points of confusion which I hope we can address relatively easily:
> >
> > 1.Diversity Problem
> >
> > The evaluation paradigm of "all possible human behaviours" is too underspecified for being useful. In Dec-POMDPs there commonly isn't a single policy that does well with all possible partners. This is one of the main challenges of Hanabi compared to  a (nearly) fully observable setting like Overcooked (outside of synchronous moves) .
> > Instead, the approach we are taking with this benchmark is pragmatic and mirrors a highly relevant real-world use case: There is a given population that an algorithm is supposed to cooperate with (*not* any arbitrary human) and it is possible to collect a _small_ dataset from this population. The task then is develop methods that allow doing well with the source population when only having access to the small dataset (in our case 1000 games).
> >
> > 2. "Cheap" proxies
> > There is a confusion here: "cheap" refers to the cost of evaluating against the human proxies, compared to running human studies. Indeed, constructing the human proxies was not "cheap" and constitutes a major contribution towards more reliable human-AI evaluation.
> >
> > Please let us know if this addresses your concerns, we will also update the paper to make sure the points come across more clearly.

---

> ### Comment · Reviewer_phEU · 2024-11-20
> **A follow up on diversity**
>
> As I work on ad hoc coordination agents, I am pretty interested in seeing a benchmark that (1) captures the diversity in evaluating partners and (2) has a diagnostic metric to tell the weakness (e.g., which cluster of conventions that the ad hoc agent cannot cooperate well).
>
> There are works that **generate** diverse evaluating teammates, like Adversity and BRDiv, but I think another way to show the diversity is to **measure** how diverse the evaluating agents are. If this work focuses more on human compatibility, then maybe it is out of my expertise.
>
> If the work is posed as an Ad Hoc teamwork problem, regarding the diversity problem, I believe that RNN could consistently capture some human trajectories and conventions within the episode; do you have a way to qualitatively show some diversity that the proxy captures human behaviors that belong to H-convention and human behaviors that do not belong to H-convention?
>
> @Reviewer 4ENk, what do you think that the authors can do to show the spectrum of the behaviors?

---

> > ### Author Response · Authors · 2024-11-23
> > **Diversity: Details & Analysis**
> >
> > We thank the reviewer for engaging in this discussion and appreciate meaningful feedback and ideas.
> >
> >  As we mentioned earlier, the task in our challenge is to develop methods that allow doing well with the source human population rather than with (arbitrary) diverse agents. We aim to mirror a real-world scenario where only a small sample of data from this population is available, and the human-centric focus holds intrinsic value. Additionally, this setting has been explored in Hanabi, but up until now, there was no way to compare and validate methods given that all human proxies and datasets stayed strictly closed source. We also stress that this is not the same setting explored in Overcooked since we focus on a partially observable setting and extend the work to three players.
> >
> > We agree that for our human proxies to be useful, they should exhibit multiple human-like strategies. We have already provided evidence of their human likeness and have now added additional behavioural analysis to the revised paper. Our findings show that the human proxies closely follow H-conventions in unrolled games. In the paper, we provide a detailed analysis of one trajectory where we find human proxies adhere to H-conventions 88% of the time. It is important to note that 100% adherence is not expected, as the H-group's guidelines state: "Even though the strategy reference outlines the 'correct' thing to do in a lot of situations, these are not hard and fast rules... Everything is flexible and can be modified by using wits, judgment, and a lot of experience." We should also note that our expertise in the game extends up to Level 4 of the conventions, so we might have missed strategies beyond this level. For further details, please see the Appendix of our paper.
> >
> > Our results are now even more significant because we have shown:
> >
> > - Strong and Robust Performance: The human proxies demonstrate strong and consistent performance.
> > - Methodology: We built our human proxies using SOTA methods, ensuring they effectively capture human-like strategies.
> > - Effective Cross-Play: When paired together, they perform well in cross-play situations, indicating convergence to similar/compatible policies.
> > - Compatibility with BC Agents: They work effectively with Behavioral Cloning (BC) agents, which are highly human-like since their policies are derived solely from human data.
> > - High Action Prediction Scores: They achieve strong scores in human-dataset action prediction tasks, with results nearly identical to those of BC agents trained exclusively on this data.
> > - Behavioural Metrics: Metrics such as Interaction Policy Prediction (IPP) and Communicativeness are almost identical between the human proxies and the human trajectories in the dataset.
> > - Diverse H-Conventions: Our analysis shows that human proxies employ diverse H-conventions in the majority of their moves, capturing a spectrum of human-like behaviours.
> >
> > We believe these findings, along with our previous responses and updates, address concerns about the diversity captured by our proxies, and their fidelity. Thank you again for your valuable feedback.
> >
> > We hope our additional analysis clarifies the strengths of our approach, and we welcome further discussion. If all their concerns are met, we ask the reviewer to consider increasing their support for our paper.

---

### Author Response · Authors · 2024-11-19
**Simplifying the Challenge**

Based on the valuable feedback from the reviewers, we realised there is an opportunity to improve and simplify the challenge to lower the entry barrier for participants. We observed that in the 5,000-game limit setting, the baseline agents already achieve strong scores, particularly in the two-player configuration. Therefore, we have decided to focus solely on the 1,000-game limit challenge

By simplifying the challenge in this way, we make it more approachable for participants. With only a single dataset to consider, candidates can focus their efforts more effectively. Additionally, this reduces the number of agents that need to be trained to participate in the challenge, making it less resource-intensive and more accessible to a wider audience. We believe this change enhances the challenge, encouraging more innovative solutions in the limited data setting.

---

> ### Comment · Reviewer_4ENk · 2024-11-20
> **About Lower the Entry Barrier**
>
> Without providing relevant baseline implementations and sufficiently diverse evaluation partners, it cannot be claimed that the barrier to entry has been lowered.

---

### Author Response · Authors · 2024-11-27
**Baseline Updates**

We would like to provide an update on our paper based on the valuable feedback we have received, given that the paper revision period is ending soon:

- **Inclusion of Other-Play Baseline**: We have updated the manuscript to include the results of the Other-Play baseline for the three-player setting.

- **Upcoming FCP Results**: We are currently training the Fictitious Co-Play (FCP) method, which involves a large population of agents and requires substantial computational time in the Hanabi environment. We promise to include FCP results in the camera-ready version of the paper.

We appreciate constructive comments, which have significantly contributed to improving our work.

---

### Author Response · Authors · 2024-12-03
**Final Reflection**

As the discussion period comes to a close, we wanted to reach out and thank all of you who engaged with us about our paper. Your insights and questions have been incredibly valuable, and we've worked hard to address all the concerns you've raised.

We have provided additional analysis, included new baselines, and offered explanations to clarify any points of confusion. While we're a bit disappointed that we didn't receive further feedback from some reviewers after making these updates, we hope that our efforts reflect our commitment to improving the paper.

We genuinely hope you'll consider our revisions and explanations when finalizing your evaluations. Your feedback has helped us enhance our work, and we appreciate the time you've invested in reviewing our submission.

Thank you again for your thoughtful contributions.

---

### Meta-Review · Area_Chair_wSdm · 2024-12-17

**Metareview:**

This paper proposes to evaluate human-AI coordination in Hanabi using human-like proxy agents. However, the authors did not provide enough motivation for solving such a task in their proposed settings. The core idea is to imitate human data, which is straightforward and widely used. While novelty along is not an issue, the lack of innovation with unclear presentation make this paper below the bar of ICLR.

**Additional Comments On Reviewer Discussion:**

The authors only partially resolved concerns during the rebuttal.

---

### Decision · Program_Chairs · 2025-01-22

Reject